# Dequantified Diffusion-Schrödinger Bridge for Density Ratio Estimation

**Wei Chen** [1]   **Shigui Li** [1]   **Jiacheng Li** [2]   **Junmei Yang** [2]   **John Paisley** [3]   **Delu Zeng** [2]

## Abstract

Density ratio estimation is fundamental to tasks involving $f$-divergences, yet existing methods often fail under significantly different distributions or inadequately overlapping supports — the *density-chasm* and the *support-chasm* problems. Additionally, prior approaches yield divergent time scores near boundaries, leading to instability. We design $\mathbf{D^3RE}$, a unified framework for **robust**, **stable** and **efficient** density ratio estimation. We propose the *dequantified diffusion bridge interpolant* (**DDBI**), which expands support coverage and stabilizes time scores via diffusion bridges and Gaussian dequantization. Building on DDBI, the proposed *dequantified Schrödinger bridge interpolant* (**DSBI**) incorporates optimal transport to solve the Schrödinger bridge problem, enhancing accuracy and efficiency. Our method offers uniform approximation and bounded time scores in theory, and outperforms baselines empirically in mutual information and density estimation tasks. Code is available at https://github.com/Hoemr/Dequantified-Diffusion-Bridge-Density-Ratio-Estimation.git.

## 1. Introduction

Quantifying distributional discrepancies via $f$-divergences-defined through density ratios $r(\mathbf{x}) = q_1(\mathbf{x})/q_0(\mathbf{x})$ is foundational in tasks such as domain adaptation, generative modeling, and hypothesis testing. However, directly estimating $r(\mathbf{x})$ by modeling $q_0$ and $q_1$ becomes intractable in high dimensions, motivating density ratio estimation (DRE) methods that bypass explicit density modeling (Sugiyama et al., 2012). While DRE underpins modern techniques like

mutual information estimation (Colombo et al., 2021) and likelihood-free inference (Thomas et al., 2022), it struggles with a critical challenge known as the *density-chasm* problem, where multi-modal or divergent distributions lead to unstable ratio estimates (Rhodes et al., 2020).

Existing methods like telescoping ratio estimation (TRE) (Rhodes et al., 2020) and its continuous extension DRE-$\infty$ (Choi et al., 2022) estimate density ratios via intermediate steps. TRE improves accuracy by adding more intermediate variables, but increases model complexity linearly. DRE-$\infty$ uses continuous-time score matching to avoid this, yet both face a core challenge in the *support-chasm problem* (see Definition 3.1), where $\mathsf{supp}(q_0) \cap \mathsf{supp}(q_1)$ is small or empty. This leads to inadequately overlapping supports and ill-defined ratios (Srivastava et al., 2023).

To address the support-chasm problem, we unify the interpolation strategies in prior works as *deterministic interpolants* (**DI**) and propose the *diffusion bridge interpolant* (**DBI**), which uses diffusion bridges to enable diverse trajectory exploration and smooth transitions between distributions. By Theorem 3.2 and Corollary 3.3, DBI expands support coverage and trajectory sets beyond existing approaches.

A second challenge arises in prior methods: As $t \to 1^-$, the absolute time score $\mathbb{E}_{q_t}[|\partial_t \log q_t|]$ diverges for both DI and DBI (Theorem 3.4), leading to unstable estimations at the boundary. To mitigate this, we propose *Gaussian dequantization* (**GD**), which addresses boundary densities $q_0$ and $q_1$ via Gaussian convolution, ensuring $\mathbb{E}_{q_t}[|\partial_t \log q_t|]$ remains bounded over $t \in [0, 1]$ (Corollary 3.8). The resulting *dequantified diffusion bridge interpolant* (**DDBI**) balances robustness and computational efficiency.

To further reduce estimation error and improve efficiency, the *dequantified Schrödinger bridge interpolant* (**DSBI**) is derived by integrating DDBI with *optimal transport rearrangement* (**OTR**), solving the Schrödinger bridge problem (Proposition 3.6). Together, applying DDBI and DSBI to DRE leads to the ***Dequantified Diffusion Bridge Density Ratio Estimation*** ($\mathbf{D^3RE}$) framework. We summarize these developments in Table 1.

Experimental results show that $D^3RE$ improves robustness and efficiency in downstream tasks such as density ratio estimation, mutual information estimation, and density es-

---

[1]The School of Mathematics, South China University of Technology, Guangzhou 510006, China [2]The School of Electronic and Information Engineering, South China University of Technology, Guangzhou 510006, China [3]The Department of Electrical Engineering, Columbia University, New York, NY 10027, USA. Correspondence to: Delu Zeng <dlzeng@scut.edu.cn>.

*Proceedings of the 42$^{nd}$ International Conference on Machine Learning*, Vancouver, Canada. PMLR 267, 2025. Copyright 2025 by the author(s).

Table 1. Comparison of advantages of interpolants in this work.

| | Diffusion bridge (robust & stable) | GD (stable) | OTR (efficient) |
|---|---|---|---|
| **DI** (previous) | | | |
| **DBI** (ours) | ✓ | | |
| **DDBI** (ours) | ✓ | ✓ | |
| **DSBI** (ours) | ✓ | ✓ | ✓ |

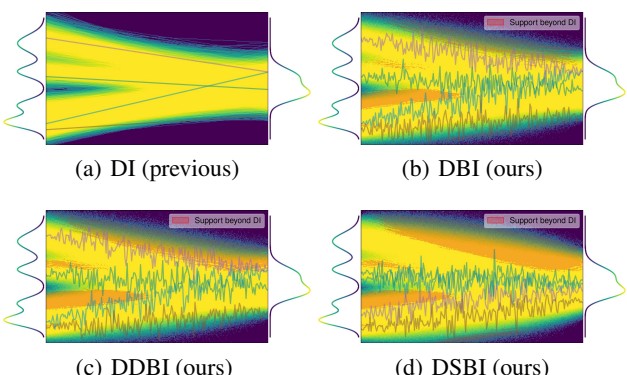

(a) DI (previous)          (b) DBI (ours)

(c) DDBI (ours)           (d) DSBI (ours)

Figure 1. Trajectory sets comparison among DI, DBI, DDBI and DSBI. Our methods yield ***broader trajectory sets***, with intermediate distributions exhibiting wider support than those of DI. The entropically regularized transport losses for subfigures (a-d) are 44.17, 31.14, 31.15, and 8.26, respectively (See Eq. (17) for details.). Lower loss indicates increased path diversity.

timation. Fig. 1 illustrates a comparison of interpolation strategies among DI, DBI, DDBI, and DSBI, with light blue points representing intermediate samples drawn from $q_0$ and $q_1$. Our proposed methods (DBI, DDBI, and DSBI) enable broader exploration of alternative trajectories, producing intermediate distributions with larger support compared to DI, consistent with Theorem 3.2 and Corollary 3.3 below.

The main contributions of this work are as follows:

- We propose D³RE, the first unified framework to address both the density-chasm and support-chasm problems via uniformly approximated density ratio estimation (Proposition 3.5). Our interpolants expand both the support (Theorem 3.2) and trajectory sets (Corollary 3.3), alleviating the support-chasm problem.
- We incorporate guidance mechanisms to improve interpolant quality: GD improves the stability of time score functions at boundary (Theorem 3.4, Corollary 3.8), while OTR leads to more stable (Theorem 3.7) and efficient (Fig. 8) estimation.
- Experiments demonstrate D³RE's superiority in density ratio estimation, mutual information estimation, and density estimation.

## 2. Background

Let $\mathbf{X}_0 \sim q_0(\mathbf{x})$ and $\mathbf{X}_1 \sim q_1(\mathbf{x})$ be random variables in $\mathbb{R}^d$ with set-theoretic supports $\mathsf{supp}(q_0)$ and $\mathsf{supp}(q_1)$. Upper cases $\mathbf{X}_0$ and $\mathbf{X}_1$ denote random variable, while lower cases $\mathbf{x}_0$ and $\mathbf{x}_1$ denote the samples of random variables.

### 2.1. Density Ratio Estimation

Density ratio estimation (DRE) aims to estimate the density ratio $r^\star(\mathbf{x}) = \frac{q_1(\mathbf{x})}{q_0(\mathbf{x})}$ using i.i.d. samples from both distributions. A common approach is density ratio matching via Bregman divergence minimization (Sugiyama et al., 2012), which optimizes a parameterized ratio $r_{\boldsymbol{\theta}}$ by minimizing

$$\mathrm{BD}(r_{\boldsymbol{\theta}}) = -\mathbb{E}_{\substack{q_0(\mathbf{x}_0) \\ q_1(\mathbf{x}_1)}} \left[ \log \frac{1}{1+r_{\boldsymbol{\theta}}(\mathbf{x}_0)} + \log \frac{r_{\boldsymbol{\theta}}(\mathbf{x}_1)}{1+r_{\boldsymbol{\theta}}(\mathbf{x}_1)} \right], \quad (1)$$

where $\boldsymbol{\theta}$ denotes the trainable parameters of $r_{\boldsymbol{\theta}}$. The minimizer of Eq. (1) satisfies $r_{\boldsymbol{\theta}^\star} = r^\star$. However, DRE suffers from the density-chasm problem (Rhodes et al., 2020).

TRE mitigates this by employing a *divide-and-conquer* strategy. It partitions the interval $[0, 1]$ into $M \in \mathbb{Z}_+$ subintervals with endpoints $\{m/M\}_{m=0}^M$, constructing intermediate variables $\mathbf{X}_{m/M} \sim q_{m/M}(\mathbf{x})$ via linear interpolation

$$\mathbf{X}_{m/M} = \sqrt{1 - \eta_{m/M}^2}\, \mathbf{X}_0 + \eta_{m/M} \mathbf{X}_1, \quad (2)$$

where $\eta_{m/M}$ increases from 0 to 1. The density ratio decomposes into a telescoping product

$$r^\star(\mathbf{x}) = \frac{q_1(\mathbf{x})}{q_0(\mathbf{x})} = \prod_{m=0}^{M-1} \frac{q_{(m+1)/M}(\mathbf{x})}{q_{m/M}(\mathbf{x})}$$
$$= \prod_{m=0}^{M-1} r_{m/M}^\star(\mathbf{x}) \approx \prod_{m=0}^{M-1} r_{\boldsymbol{\theta}_m}(\mathbf{x}), \quad (3)$$

where $r_{m/M}^\star$ is the target intermediate density ratio, estimated by a parameterized neural network $r_{\boldsymbol{\theta}_m}$ with trainable parameters $\boldsymbol{\theta}_m$. In this case, $M$ networks must be trained. While a larger $M$ improves accuracy, it increases computational cost and may still fail to sufficiently reduce the KL divergence, $\mathrm{KL}(q_{m/M} \| q_{(m+1)/M})$, leaving the density-chasm problem unaddressed.

DRE-$\infty$ (Choi et al., 2022) extends TRE to $M \to \infty$, representing the log ratio as an integral of the time score

$$\log r^\star(\mathbf{x}) = \int_0^1 \partial_t \log q_t(\mathbf{x}) \mathrm{d}t \approx \int_0^1 s_{\boldsymbol{\theta}^\star}^t(\mathbf{x}, t) \mathrm{d}t, \quad (4)$$

where $\partial_t \log q_t(\mathbf{x}) \approx (\log q_{t+\Delta t}(\mathbf{x}) - \log q_t(\mathbf{x}))/\Delta t$ with an infinitesimal gap $\Delta t = \lim_{M \to \infty} 1/M$ denotes the time score. The time score model $s_{\boldsymbol{\theta}}^t$ approximates the time score via minimization of

$$\mathcal{L}_1 = \mathbb{E}_{q(t)q_t(\mathbf{x})} \left[ \lambda(t) \left| \partial_t \log q_t(\mathbf{x}) - s_{\boldsymbol{\theta}}^t(\mathbf{x}, t) \right|^2 \right], \quad (5)$$

where $\lambda(\cdot) : [0,1] \to \mathbb{R}_+$ is a time-dependent weighting function, and $q(t) = \mathcal{U}[0,1]$ is the uniform distribution over the interval [0,1]. When the time score model satisfies $\partial_t \log q_t(\mathbf{x}) = s_{\boldsymbol{\theta}^\star}^t(\mathbf{x}, t)$, the loss function reaches its minimum value $\mathcal{L}_1(\boldsymbol{\theta}^\star)$. Unlike in TRE, only one network, $s_{\boldsymbol{\theta}}^t$, needs to be trained in DRE-$\infty$.

## 2.2. Denoising Diffusion Bridge Model

Denoising diffusion models (DDMs) simulate a diffusion process $\{\mathbf{X}_t\}_{t \in [0,1]}$, which serves as a continuous bridge between $\mathbf{X}_0$ and $\mathbf{X}_1$. This process is described by the solution to an Itô stochastic differential equation (SDE)

$$d\mathbf{X}_t = \mathbf{f}(\mathbf{X}_t, t)dt + g(t)d\mathbf{W}_t, \qquad (6)$$

where $\{\mathbf{W}_t\}_{t \in [0,1]}$ is a standard Wiener process, $\mathbf{f} : \mathbb{R}^d \times [0,1] \to \mathbb{R}^d$ and $g : [0,1] \to \mathbb{R}$ are termed as the drift coefficient and scalar diffusion coefficient of $\mathbf{X}_t$, respectively.

Conventional DDMs (Li et al., 2023; Song et al., 2021; Li et al., 2024a; Xu et al., 2024) require either $q_0$ or $q_1$ to be a simple, tractable distribution (e.g., isotropic Gaussian), which limits their ability to bridge arbitrary complex distributions and restricts applications such as DRE.

Denoising diffusion bridge models (Zhou et al., 2024a) overcome this limitation by simulating stochastic processes that interpolate between paired distributions with $\mathbf{X}_1$ as endpoints. These processes are derived from the SDE in Eq. (6) via Doob's $h$-transform (Doob & Doob, 1984),

$$d\mathbf{X}_t = [\mathbf{f}(\mathbf{X}_t, t) + g^2(t)\mathbf{h}(\mathbf{X}_t, t, \mathbf{X}_1)]dt + g(t)d\mathbf{W}_t, \quad (7)$$

where $\mathbf{h}(\mathbf{X}_t, t, \mathbf{X}_1) = \nabla_{\mathbf{x}_t} \log p_t(\mathbf{X}_1 \mid \mathbf{X}_t)$ is the $h$ function representing the gradient of the log transition kernel from time $t$ to 1. The process explicitly depends on $\mathbf{X}_1$ and, given $\mathbf{X}_0$ and $\mathbf{X}_1$, it forms a *diffusion bridge* with transition kernel $q_t(\mathbf{x} \mid \mathbf{X}_0, \mathbf{X}_1)$. A special case of the diffusion bridge, termed the *Brownian bridge*, arises under the conditions $\mathbf{f}(\mathbf{X}_t, t) := 0, g(t) := 1$ and $\mathbf{h}(\mathbf{X}_t, t, \mathbf{X}_1) := \frac{\mathbf{X}_1 - \mathbf{X}_t}{1-t}$. The Brownian bridge is defined as the solution to $d\mathbf{B}_t = \frac{\mathbf{B}_1 - \mathbf{B}_t}{1-t}dt + d\mathbf{W}_t$ and its transition kernel is given by $q_t(\mathbf{b} \mid \mathbf{B}_0, \mathbf{B}_1) = \mathcal{N}((1-t)\mathbf{B}_0 + t\mathbf{B}_1, t(1-t)\mathbf{E}_d)$.

# 3. Method

In this section, we we extend prior works into a unified framework called **D**equantified **D**iffusion-bridge **D**ensity-**R**atio **E**stimation (**D**$^3$**RE**). D$^3$RE offers a robust and efficient solution for density ratio estimation, theoretically mitigating the density-chasm and support-chasm problems.

## 3.1. Support-chasm Problem

**Deterministic Interpolant.** We summarize the interpolants used in prior works, such as TRE and DRE-$\infty$,

to *deterministic interpolant* (**DI**), defined as

$$\mathbf{I}(\mathbf{X}_0, \mathbf{X}_1, t) = \alpha_t \mathbf{X}_0 + \beta_t \mathbf{X}_1, \qquad (8)$$

where $\mathbf{I} : \mathbb{R}^d \times \mathbb{R}^d \times [0,1] \to \mathbb{R}^d$ is an interpolant continuously differentiable in $(\mathbf{X}_0, \mathbf{X}_1, t)$ and the time-indexed coefficients $\alpha_t, \beta_t \in C^2[0,1]$ are monotonic with $\alpha_t$ decreasing and $\beta_t$ increasing in $t$. They are strictly positive and satisfy boundary conditions $\alpha_0 = \beta_1 = 1$ and $\alpha_1 = \beta_0 = 0$, with constraints $\alpha_t + \beta_t = 1$ or $\alpha_t^2 + \beta_t^2 = 1, \forall t \in [0,1]$.

Prior methods, such as TRE (Rhodes et al., 2020) and DRE-$\infty$ (Choi et al., 2022), are specific cases of DI distinguished by their choices of $\alpha_t$ and $\beta_t$ (see Appendix B.1).

However, DRE with DI suffers from the support-chasm problem, where minimal overlap between $\mathsf{supp}(q_0)$ and $\mathsf{supp}(q_1)$ leads to ill-defined ratios (Srivastava et al., 2023).

**Definition 3.1** (Support-chasm Problem). Let $q_0, q_1$ be probability density functions with supports $\mathsf{supp}(q_0)$ and $\mathsf{supp}(q_1)$, respectively. For a given threshold $\varepsilon > 0$, if $\mu(\mathsf{supp}(q_0) \cap \mathsf{supp}(q_1)) < \varepsilon$, then a *support-chasm* exists between $q_0$ and $q_1$, where $\mu$ is the Lebesgue measure.

**Diffusion Bridge Interpolant.** To mitigate the support-chasm, we introduce a Brownian bridge $\{\mathbf{B}_t\}_{t \in [0,1]}$, leading to the *diffusion bridge interpolant* (**DBI**)

$$\mathbf{X}_t = \mathbf{I}(\mathbf{X}_0, \mathbf{X}_1, t) + \gamma \mathbf{B}_t, \qquad (9)$$

where $\gamma \in \mathbb{R}_{\geq 0}$ is the noise factor controlling the stochastic component $\mathbf{B}_t$. This factor provides flexibility by adjusting the variability introduced by the bridge at different stages of interpolation. *When $\gamma = 0$, DBI reduces to DI.*

Since $\mathbf{B}_t$ is a Gaussian process with zero mean and variance $t(1-t)$, i.e., $\mathbf{B}_t \sim \mathcal{N}(\mathbf{0}, t(1-t)\mathbf{E}_d)$, the transition kernel of the DBI can be derived follows from Eq. (9) as $q_t(\mathbf{x} \mid \mathbf{X}_0, \mathbf{X}_1) = \mathcal{N}(\mathbf{I}(\mathbf{X}_0, \mathbf{X}_1, t), t(1-t)\gamma^2 \mathbf{E}_d)$. By applying the reparameterization trick, DBI admits the equivalent form

$$\mathbf{X}_t = \mathbf{I}(\mathbf{X}_0, \mathbf{X}_1, t) + \sqrt{t(1-t)\gamma^2}\mathbf{Z}_t, \qquad (10)$$

where $\mathbf{Z}_t \sim \mathcal{N}(\mathbf{0}, \mathbf{E}_d)$, ensuring analytical tractability and efficient implementation. The term $\sqrt{t(1-t)\gamma^2}\mathbf{Z}_t$ adds controlled variability and provides robustness and flexibility to the interpolant, expanding the support of $q_t$.

**Theorem 3.2** (Support Set Expansion). *Let $\mathbf{X}_0$ and $\mathbf{X}_1$ be random variables. Let $q_t(\mathbf{x})$ and $q_t'(\mathbf{x})$ denote the marginal densities under DI and DBI, respectively. Then, for any $t \in (0,1)$, the support of $q_t'$ includes or expands beyond the support of $q_t$, i.e., $\mathsf{supp}(q_t') \supseteq \mathsf{supp}(q_t)$.*

See Appendix A.1 for detailed derivation. This result shows that DBI covers a larger or equal region of the space compared to DI, providing theoretical justification for its ability to mitigate the support-chasm problem in D$^3$RE.

**Corollary 3.3** (Trajectory Set Expansion). *Under the same setup as in Theorem 3.2, let the trajectory sets generated by the DI and the DBI be denoted by $\mathbb{T} = \{\{\mathbf{x}_t\}_{t\in[0,1]}; \mathbf{x}_t \in \mathrm{supp}(q_t)\}$ and $\mathbb{T}' = \{\{\mathbf{x}'_t\}_{t\in[0,1]}; \mathbf{x}'_t \in \mathrm{supp}(q'_t)\}$, respectively. Then, $\mathbb{T}'$ contains $\mathbb{T}$, i.e., $\mathbb{T}' \supseteq \mathbb{T}$.*

See Appendix A.2 for details. This generalizes the support expansion result to entire paths, implying that DBI, by injecting noise, explores a broader set of trajectories than DI. As a result, it provides better coverage of the interpolation space, enhancing robustness across diverse distributions.

### 3.2. Gaussian Dequantization for Mollified Time Score

A second fundamental challenge in conventional interpolants such as DI and DBI is the divergence of the absolute time score, as shown in Theorem 3.4.

**Theorem 3.4.** *Let $\{\mathbf{X}_t\}_{t\in[0,1]}$ be a DI defined in Eq. (8). Under Assumption A.1 and Assumption A.2, the time score for any $t \in (0,1)$ satisfies*

$$\mathbb{E}_{q_t}[|\partial_t \log q_t|] \geq d\left((1-L)\frac{|\dot{\alpha}_t|}{\alpha_t} - L\frac{|\dot{\beta}_t|}{\beta_t}\right) \\ - \mathcal{O}\left(\sqrt{\mathbb{E}_{q_t}[\|\nabla \log q_t\|^2]}\right), \quad (11)$$

*where $L$ is the Lipschitz constant in Assumption A.2. Moreover, if $L < 1$, this lower bound diverges to infinity*

$$\lim_{t\to 0^+} \mathbb{E}_{q_t}[|\partial_t \log q_t|] = -\infty, \\ \lim_{t\to 1^-} \mathbb{E}_{q_t}[|\partial_t \log q_t|] = +\infty. \quad (12)$$

Proofs can be found in Appendix A.3.

**Dequantified Diffusion Bridge Interpolant.** To stabilize the time score near the boundary, we introduce Gaussian dequantization (GD) by adding controlled perturbations to boundary samples, effectively handling the boundary densities and resulting uniformly bounded time score across $[0,1]$ (see Corollary 3.8 for details).

Specifically, for $\mathbf{x}_i \sim q_i$, its dequantified form $\mathbf{x}'_i$ can be obtained by

$$\mathbf{x}'_i = \mathbf{x}_i + \mathbf{z}_\varepsilon, \quad \mathbf{z}_\varepsilon \sim \mathcal{N}(\mathbf{0}, \varepsilon\mathbf{E}_d), \quad i \in \{0,1\}, \quad (13)$$

where $\varepsilon \in \mathbb{R}_+$ is small. The resulting dequantified densities are obtained via Gaussian convolution, $q'_i = q_i * \mathcal{N}(\mathbf{0}, \varepsilon\mathbf{E}_d)$. This smoothing ensures bounded time scores near $t = 0$ and $t = 1$ (see Theorem 3.7 and Corollary 3.8 for details), improving stability in DRE.

Incorporating GD into the DBI yields the *dequantified diffusion bridge interpolant* (**DDBI**), formulated as

$$\mathbf{X}'_t = \mathbf{I}(\mathbf{X}'_0, \mathbf{X}'_1, t) + \sqrt{t(1-t)\gamma^2}\mathbf{Z}_t. \quad (14)$$

The DDBI can be expressed in terms of the original DBI by defining perturbed variables as $\mathbf{X}'_i = \mathbf{X}_i + \mathbf{Z}_\varepsilon$, where $\mathbf{Z}_\varepsilon \sim \mathcal{N}(\mathbf{0}, \varepsilon\mathbf{E}_d)$. This results in

$$\mathbf{X}'_t = \mathbf{I}(\mathbf{X}_0, \mathbf{X}_1, t) + \sqrt{t(1-t)\gamma^2 + (\alpha_t^2 + \beta_t^2)\varepsilon}\mathbf{Z}_t. \quad (15)$$

Here, the additional term $(\alpha_t^2 + \beta_t^2)\varepsilon$ reflects the effect of GD, yielding smoother interpolation. As a result, the transition kernel of the DDBI, $q'_t(\mathbf{x} \mid \mathbf{x}_0, \mathbf{x}_1)$, is given by $\mathcal{N}(\mathbf{I}(\mathbf{x}_0, \mathbf{x}_1, t), (t(1-t)\gamma^2 + (\alpha_t^2 + \beta_t^2)\varepsilon)\mathbf{E}_d)$, which shares the same mean trajectory as DBI (see Eq. (10)).

We have also analyzed the uniform approximation of density ratio using the DDBI. The relationship between $r(\mathbf{x})$ and $r'(\mathbf{x}) = \frac{q'_1(\mathbf{x})}{q'_0(\mathbf{x})}$ is characterized by Proposition 3.5.

**Proposition 3.5.** *Let $r(\mathbf{x})$ and $r'(\mathbf{x})$ be the density ratios with and without GD, respectively. Then, $r'$ is a uniform apporximation of $r$, with the error bounded by:*

$$\|r' - r\|_{L^\infty} \leq \mathcal{O}(\varepsilon). \quad (16)$$

*Thus, as $\varepsilon \to 0$, $r'(\mathbf{x}) \to r(\mathbf{x})$ in the uniform norm.*

See Appendix A.4 for detailed derivation. Proposition 3.5 confirms that the dequantified density ratio $r'(\mathbf{x})$ is a *uniform approximation* of $r(\mathbf{x})$ for sufficiently small $\varepsilon$.

### 3.3. Optimal Transport Rearrangement

To further reduce estimation error and improve efficiency of DRE, we extend the probability path of DDBI by aligning it with the entropically regularized optimal transport (OT)

$$\pi_{2\gamma^2} = \underset{\pi \in \Pi(q'_0, q'_1)}{\mathrm{argmin}} \int \|\mathbf{x}_0 - \mathbf{x}_1\|^2 d\pi - 2\gamma^2 \mathcal{H}(\pi), \quad (17)$$

where $\Pi(q'_0, q'_1)$ is the set of all probability paths with marginals $q'_0$ and $q'_1$, and $\mathcal{H}(\pi)$ is the entropy of $\pi$. The coefficient $2\gamma^2$ is regularization factor (details in Appendix B.2).

We apply the scalable Sinkhorn algorithm (Cuturi, 2013) to mini-batches $\{(\mathbf{x}'_0, \mathbf{x}'_1)_n\}_{n=1}^N \sim q'_0 \times q'_1$, obtaining rearranged sample pairs $\{(\hat{\mathbf{x}}'_0, \hat{\mathbf{x}}'_1)_n\}_{n=1}^N \sim \pi_{2\gamma^2}$. For convenience, we refer to this procedure as *optimal transport rearrangement* (OTR), which preserves the marginals and only changes the sample pairing.

**Dequantified Schrödinger Bridge Interpolant.** Applying OTR followed by DDBI yields the *dequantified Schrödinger bridge interpolant* (**DSBI**)

$$\hat{\mathbf{X}}'_t = \mathbf{I}(\hat{\mathbf{X}}'_0, \hat{\mathbf{X}}'_1, t) + \sqrt{t(1-t)\gamma^2 + (\alpha_t^2 + \beta_t^2)\varepsilon}\mathbf{Z}_t, \quad (18)$$

where $\alpha_t = 1 - t$ and $\beta_t = t$ are fixed.

We show that rearranging the mini-batches via OTR leads to an interpolant that naturally solves the Schrödinger bridge (SB) problem (Schrödinger, 1932).

**Proposition 3.6.** *The probability path defined by DSBI solves the SB problem*

$$\pi^\star = \underset{\pi \in \Pi(q_0', q_1')}{\arg\min} \ \mathrm{KL}(\pi \| \pi_{\mathrm{ref}}), \qquad (19)$$

*where $\pi_{\mathrm{ref}}$ is a Wiener process scaled by $\gamma$.*

See Appendix A.5 for proof. This result suggests that DSBI, as a principled integration of DDBI and OTR, implicitly solves the SB problem and provides a minimum-cost stochastic interpolation between $q_0'$ and $q_1'$.

Furthermore, to rigorously quantify the improvement brought by OTR, we establish the following result comparing the upper error bounds of DDBI and DSBI.

**Theorem 3.7.** *Consider the DDBI and DSBI with $\alpha_t = 1-t$, $\beta_t = t$. Let $\pi \in \Pi(q_0', q_1')$ be any coupling for DDBI, and $\pi_{2\gamma^2}$ the entropically regularized OT coupling for DSBI. Then, for all $t \in [0, 1]$, the variance of the time score under DSBI is no greater than that under DSBI, i.e.,*

$$\mathrm{Var}_{q_t'}^{DSBI}(\partial_t \log q_t') \leq \mathrm{Var}_{q_t'}^{DDBI}(\partial_t \log q_t'). \qquad (20)$$

**Corollary 3.8.** *For all $t \in [0, 1]$, the time score of DDBI is uniformly bounded by*

$$\mathbb{E}_{q_t'}[|\partial_t \log q_t'|] \leq \sqrt{\frac{1}{\sigma_t^2}\mathbb{E}_\pi\left[\|\mathbf{X}_0' - \mathbf{X}_1'\|^2\right] + \frac{\dot{\sigma}_t^4 d}{2\sigma_t^4}}, \quad (21)$$

*where $\sigma_t^2 = t(1-t)\gamma^2 + (2t^2 - 2t + 1)\varepsilon$ is strictly positive.*

See Appendix A.6 and Appendix A.7 for detailed derivations of Theorem 3.7 and Corollary 3.8. These results establish that both DDBI and DSBI admit a bounded time-integrated time score, $\mathbb{E}_{q_t'}[|\partial_t \log q_t'|]$, in contrast to the divergent lower bound exhibited by DI (see Theorem 3.4). Moreover, it provides a formal justification for the error reduction achieved by DSBI through the coupling $\pi_{2\gamma^2}$.

### 3.4. Dequantified Diffusion Bridge DRE

For the DDBI, $r'(\mathbf{x})$ can be approximated effectively using a neural network, as formulated in Definition 3.9. See Appendix A.8 for detailed derivation.

**Definition 3.9** (Dequantified Diffusion bridge Density Ratio Estimation, $D^3$RE)**.** Given the marginal probability density of DDBI, $q_t'(\mathbf{x})$, the log density ratio for a given point $\mathbf{x} \in \mathbb{R}^d$ can be estimated as

$$\log r^\star(\mathbf{x}) \approx \int_0^1 \partial_t \log q_t'(\mathbf{x}) \mathrm{d}t, \qquad (22)$$

where $\partial_t \cdot$ denotes the time derivative operator.

**Time Score-matching Loss.** We train a time score model $s_{\boldsymbol{\theta}}^t(\mathbf{x}, t)$ to approximate the time score $\partial_t \log q_t'(\mathbf{x})$ by minimizing the time score-matching loss (Choi et al., 2022),

$$\mathcal{L}_2 = \mathbb{E}_{q(t)q_t'(\mathbf{x})}\left[\lambda(t)\left|\partial_t \log q_t'(\mathbf{x}) - s_{\boldsymbol{\theta}}^t(\mathbf{x}, t)\right|^2\right]. \quad (23)$$

However, $\partial_t \log q_t'(\mathbf{x})$ is intractable in practice. To bypass this, an equivalent integration-by-parts form (Song & Ermon, 2020; Choi et al., 2022) is proposed

$$\mathcal{L}_3 = \mathbb{E}_{q_0'(\mathbf{x}_0)q_1'(\mathbf{x}_1)}[\lambda(0)s_{\boldsymbol{\theta}}^t(\mathbf{x}_0, 0) - \lambda(1)s_{\boldsymbol{\theta}}^t(\mathbf{x}_1, 1)]$$
$$+ \mathbb{E}_{q(t)q_t'(\mathbf{x})}\left[\partial_t\left[\lambda(t)s_{\boldsymbol{\theta}}^t(\mathbf{x}, t)\right] + \frac{1}{2}\lambda(t)s_{\boldsymbol{\theta}}^t(\mathbf{x}, t)^2\right], \quad (24)$$

where $\partial_t\left[\lambda(t)s_{\boldsymbol{\theta}}^t(\mathbf{x}, t)\right] = \lambda(t)\partial_t s_{\boldsymbol{\theta}}^t(\mathbf{x}, t) + \lambda'(t)s_{\boldsymbol{\theta}}^t(\mathbf{x}, t)$, $\partial_t s_{\boldsymbol{\theta}}^t(\mathbf{x}, t)$ and $\lambda'$ denote the time derivative of the time score model and weighting function, respectively. The first two terms enforce the boundary conditions. $\mathcal{L}_2$ and $\mathcal{L}_3$ differ only by a constant $C$ independent of $\boldsymbol{\theta}$. In practice, for stable and effective training, the joint score-matching loss is implemented, as described in Appendix C.2.

**Estimating Target Log Density Ratio.** Given the optimal parameters $\boldsymbol{\theta}^\star$ obtained by minimizing $\mathcal{L}3$, the log density ratio at any point $\mathbf{x}$ can be estimated as

$$\log r^\star(\mathbf{x}) \approx \int_0^1 \partial_t \log q_t'(\mathbf{x})\mathrm{d}t \approx \int_0^1 s_{\boldsymbol{\theta}^\star}^t(\mathbf{x}, t)\mathrm{d}t, \quad (25)$$

based on Definition 3.9. See Algorithms 1 and 2 for the full training and estimation procedures using DDBI and DSBI.

---

**Algorithm 1** Training and estimation of $D^3$RE with DDBI

**Input:** Probability densities $q_0$ and $q_1$, time score model $s_{\boldsymbol{\theta}}^t$, coefficients $\alpha_t$ and $\beta_t$, noise factor $\gamma$ and $\varepsilon$.
  **Initialize:** trainable parameters $\boldsymbol{\theta}$ of $s_{\boldsymbol{\theta}}^t$, a given point $\mathbf{x}$.
  $\mathbf{x}_0 \sim q_0(\mathbf{x}), \mathbf{x}_1 \sim q_1(\mathbf{x}), t \sim \mathcal{U}(0, 1)$
  $\mathbf{z}_\varepsilon \sim \mathcal{N}(\mathbf{0}, \varepsilon\mathbf{E}_d), \mathbf{z} \sim \mathcal{N}(\mathbf{0}, \mathbf{E}_d)$
  $\mathbf{x}_0' \leftarrow \mathbf{x}_0 + \mathbf{z}_\varepsilon, \mathbf{x}_1' \leftarrow \mathbf{x}_1 + \mathbf{z}_\varepsilon$     % GD
  $\mathbf{x}_t' \leftarrow \alpha_t\mathbf{x}_0' + \beta_t\mathbf{x}_1' + \sqrt{t(1-t)\gamma^2}\mathbf{z}$     % DBI
  $\boldsymbol{\theta}^\star \leftarrow \mathrm{Adam}(\boldsymbol{\theta}, \nabla_{\boldsymbol{\theta}}\mathcal{L}_3(\boldsymbol{\theta}))$
  $\log r(\mathbf{x}) \leftarrow \mathsf{odeint\_adjoint}(s_{\boldsymbol{\theta}^\star}^t, (0, 1), \mathbf{x})$
**Output:** estimated log density ratio $\log r(\mathbf{x})$.

---

**Algorithm 2** Training and estimation of $D^3$RE with DSBI

**Input:** Probability densities $q_0$ and $q_1$, time score model $s_{\boldsymbol{\theta}}^t$, coefficients $\alpha_t$ and $\beta_t$, noise factor $\gamma$ and $\varepsilon$.
  **Initialize:** trainable parameters $\boldsymbol{\theta}$ of $s_{\boldsymbol{\theta}}^t$, a given point $\mathbf{x}$.
  $\mathbf{x}_0 \sim q_0(\mathbf{x}), \mathbf{x}_1 \sim q_1(\mathbf{x}), t \sim \mathcal{U}(0, 1)$
  $\mathbf{z}_\varepsilon \sim \mathcal{N}(\mathbf{0}, \varepsilon\mathbf{E}_d), \mathbf{z} \sim \mathcal{N}(\mathbf{0}, \mathbf{E}_d)$
  $\mathbf{x}_0' \leftarrow \mathbf{x}_0 + \mathbf{z}_\varepsilon, \mathbf{x}_1' \leftarrow \mathbf{x}_1 + \mathbf{z}_\varepsilon$     % GD
  $\pi_{2\gamma^2} \leftarrow \mathsf{Sinkhorn}(\mathbf{x}_0', \mathbf{x}_1', 2\gamma^2)$     % OTR
  $(\hat{\mathbf{x}}_0', \hat{\mathbf{x}}_1') \sim \pi_{2\gamma^2}$
  $\hat{\mathbf{x}}' \leftarrow \alpha_t\hat{\mathbf{x}}_0' + \beta_t\hat{\mathbf{x}}_1' + \sqrt{t(1-t)\gamma^2}\mathbf{z}$     % DBI
  $\boldsymbol{\theta}^\star \leftarrow \mathrm{Adam}(\boldsymbol{\theta}, \nabla_{\boldsymbol{\theta}}\mathcal{L}_3(\boldsymbol{\theta}))$
  $\log r(\mathbf{x}) \leftarrow \mathsf{odeint\_adjoint}(s_{\boldsymbol{\theta}^\star}^t, (0, 1), \mathbf{x})$
**Output:** estimated log density ratio $\log r(\mathbf{x})$.

---

# 4. Experiments

For experiments involving D$^3$RE, we implement both DDBI and DSBI. Unless specified otherwise, we use the following settings: $\alpha_t = 1 - t, \beta_t = t, \gamma^2 = 0.5, \varepsilon = 1e - 5$ and $\lambda(t) = \gamma^2 t(1 - t)$. Under this configuration, the interpolant $\mathbf{I}(\mathbf{X}_0, \mathbf{X}_1, t) = (1 - t)\mathbf{X}_0 + t\mathbf{X}_1$ aligns with the Benamou-Brenier solution to the optimal transport problem in Euclidean space (McCann, 1997). The parameterized score model is trained with time score matching loss $\mathcal{L}_3$ and optimized with Adam optimization method.

## 4.1. Density Estimation

Let $r(\mathbf{x}) = \frac{q_1(\mathbf{x})}{q_0(\mathbf{x})}$ be the target density ratio, where $q_1(\mathbf{x})$ is an intractable data distribution, and $q_0(\mathbf{x})$ is the simpler, tractable noise distribution. Once the estimated density ratio $r_{\boldsymbol{\theta}^\star}$ is obtained, the log-density of $q_1$ can be approximated as $\log q_1(\mathbf{x}) \approx \log r_{\boldsymbol{\theta}^\star}(\mathbf{x}) + \log q_0(\mathbf{x})$.

**2-D Synthetic Datasets.** We trained DRE-$\infty$ (baseline) and D$^3$RE (ours) on eight 2-D synthetic datasets, including swissroll, circles, rings, moons, 8gaussians, pinwheel, 2spirals, and checkerboard, for 20,000 epochs using the joint score matching loss (details in Appendix C.2). The density estimation results are shown in Fig. 2.

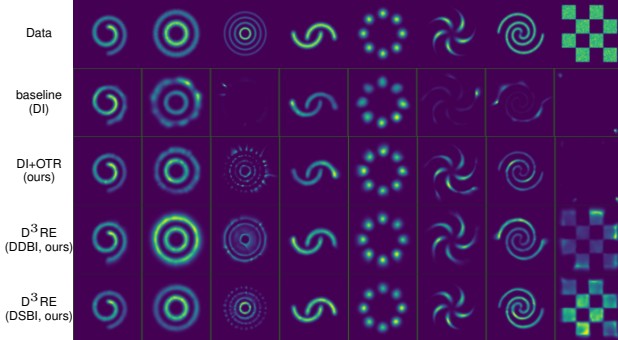

*Figure 2.* Density estimation results on eight 2-D synthetic datasets for different methods. D$^3$RE effectively estimates the density for both multi-modal and discontinuous distributions.

Our experiments demonstrate that D$^3$RE effectively estimates the density for both multi-modal and discontinuous distributions, outperforming DRE-$\infty$. The baseline struggles, especially with complex datasets like rings and checkerboard, where significant distortions occur. While the OTR trick reduces errors, it remains insufficient for intricate datasets like 2spirals and pinwheel.

In contrast, the D$^3$RE framework, leveraging both DDBI and DSBI, achieves significantly improved density estimation. DDBI captures fine-grained details, while DSBI provides robust performance across all datasets, particularly excelling in modeling complex distributions and mitigating

estimation artifacts. Further comparisons across training epochs are available in Appendix C.4.

**Energy-based Modeling on MNIST.** We applied the proposed D$^3$RE framework for density estimation on the MNIST dataset, leveraging pre-trained energy-based models (EBMs) (Choi et al., 2022). Experimental details are in Appendix C.4. The results are reported in bits-per-dimension (BPD). Results in Tab. 2 show that D$^3$RE achieves the lowest BPD values across all noise types (Gaussian, Copula, and RQ-NSF), outperforming baselines and existing methods. See also Tab. 3 in the Appendix.

Specifically, D$^3$RE consistently surpasses DRE-$\infty$ and its variant with OTR. The DSBI method delivers the best overall results, achieving BPD values of 1.293 (Gaussian), 1.170 (Copula), and 1.066 (RQ-NSF), demonstrating its robustness and effectiveness in optimizing density estimates. Compared to traditional methods like NCE and TRE, D$^3$RE shows significant improvements, especially under challenging noise distributions like Gaussian and Copula, where baseline methods yield higher BPD. These findings underscore D$^3$RE's superior performance in accurately estimating densities and modeling complex data distributions.

*Table 2.* Comparison of the estimated densities on MNIST dataset based on pre-trained energy-based models. The results are reported in bits-per-dim (BPD). Lower is better. The reported results for NCE and TRE are from Rhodes et al. (2020).

| Method | Noise type | Noise | BPD ($\downarrow$) |
|---|---|---|---|
| NCE | Gaussian | 2.01 | 1.96 |
| TRE | Gaussian | 2.01 | 1.39 |
| DRE-$\infty$ | Gaussian | 2.01 | 1.33 |
| DRE-$\infty$+OTR | Gaussian | 2.01 | 1.313 |
| D$^3$RE (DDBI) | Gaussian | 2.01 | 1.297 |
| D$^3$RE (DSBI) | Gaussian | 2.01 | **1.293** |
| NCE | Copula | 1.40 | 1.33 |
| TRE | Copula | 1.40 | 1.24 |
| DRE-$\infty$ | Copula | 1.40 | 1.21 |
| DRE-$\infty$+OTR | Copula | 1.40 | 1.204 |
| D$^3$RE (DDBI) | Copula | 1.40 | 1.193 |
| D$^3$RE (DSBI) | Copula | 1.40 | **1.170** |
| NCE | RQ-NSF | 1.12 | 1.09 |
| TRE | RQ-NSF | 1.12 | 1.09 |
| DRE-$\infty$ | RQ-NSF | 1.12 | 1.09 |
| DRE-$\infty$+OTR | RQ-NSF | 1.12 | 1.072 |
| D$^3$RE (DDBI) | RQ-NSF | 1.12 | 1.072 |
| D$^3$RE (DSBI) | RQ-NSF | 1.12 | **1.066** |

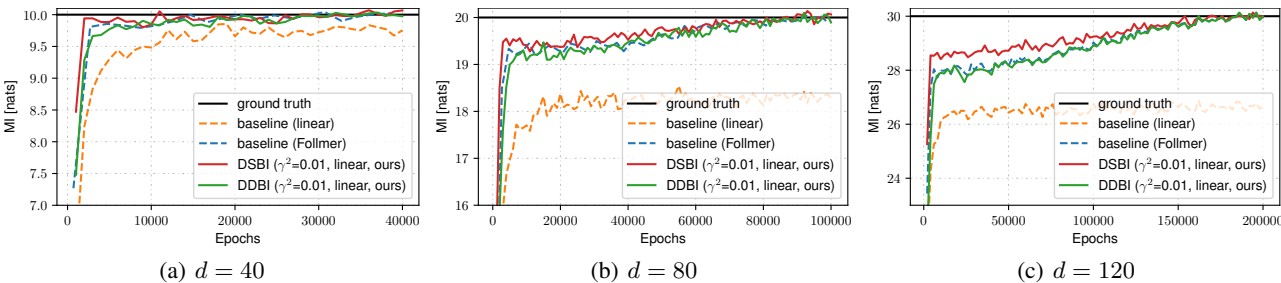

(a) $d = 40$          (b) $d = 80$          (c) $d = 120$

*Figure 3.* Evolution of estimated MI across epochs with varying methods and dimensions $d = \{40, 80, 120\}$. D³RE outperforms the baseline in both speed and precision. DRE-∞ (Choi et al., 2022) is regarded as the 'baseline' in this experiment.

## 4.2. Mutual Information Estimation

Mutual information (MI) measures the dependency between two random variables $\mathbf{X} \sim p(\mathbf{x})$ and $\mathbf{Y} \sim q(\mathbf{y})$, defined as $\mathrm{MI}(\mathbf{X}, \mathbf{Y}) = \mathbb{E}_{p(\mathbf{x}, \mathbf{y})} \left[ \log \frac{p(\mathbf{x}, \mathbf{y})}{p(\mathbf{x})q(\mathbf{y})} \right]$. Here $p(\mathbf{x}, \mathbf{y})$ be their joint density. $q(\mathbf{y}) = \mathcal{N}(\mathbf{0}, \sigma^2 \mathbf{E}_d)$ and $p(\mathbf{x}) = \mathcal{N}(\mathbf{0}, \mathbf{E}_d)$, with $\sigma^2 = 1e - 6$ and $d = \{40, 80, 120\}$, are two $d$-dimensional correlated Gaussian distributions. The experimental setup in DRE-∞ (Choi et al., 2022) is adapted to implement D³RE. DRE-∞ serves as the benchmark. More details can be found in Appendix C.3.

The evolution of estimated MI across epochs for $d = \{40, 80, 120\}$, comparing D³RE with DRE-∞, are analyzed. Results in Fig. 3 show that the red (DSBI) and green (DDBI) curves outperform the blue and yellow (DRE-∞) curves in two aspects. First, D³RE converges to the true MI value more rapidly as it expands trajectory sets (see Corollary 3.3), improving interpolation accuracy. Second, it exhibits greater stability with fewer fluctuations around the true MI, indicating more reliable estimates.

We conclude that D³RE outperforms the baseline in both speed and precision. For $d = 120$, the MI estimated by D³RE is much robust than that of DRE-∞.

## 4.3. Analysis and Discussion

**Ablation Study on $\gamma^2$.** The ablation study on $\gamma^2$ for density estimation (Fig. 4) reveals systematic trade-offs in performance across regularization strengths. For small $\gamma^2 = 0.001$, the model achieves rapid initial alignment with the ground truth distribution (first row) but exhibits overfitting artifacts in later epochs, manifesting as irregular density peaks and deviations from the smooth ground truth structure. Intermediate values ($\gamma^2 = 0.01$–$0.1$) demonstrate balanced behavior: $\gamma^2 = 0.01$ preserves finer details while maintaining stability, and $\gamma^2 = 0.1$ produces smoother approximations with minimal divergence from the true distribution. Larger $\gamma^2$ values ($\geq 0.5$) induce excessive regularization, leading to oversmoothed estimates that fail to capture crit-

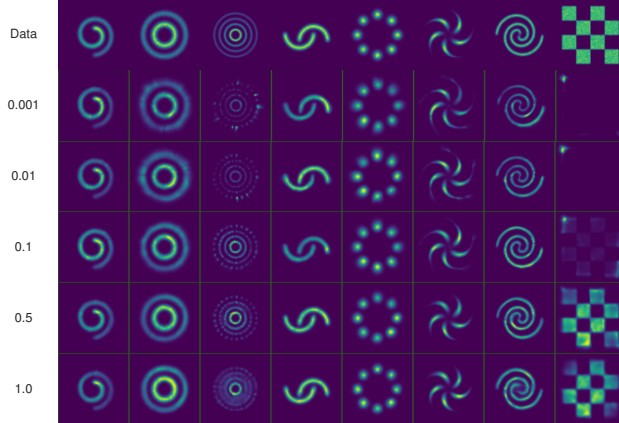

*Figure 4.* Ablation study on the effect of $\gamma^2$ for density estimation on 2-D toy data. The first row displays the results for the ground truth data. Each subsequent row, from top to bottom, corresponds to $\gamma^2$ values of 0.001, 0.01, 0.1, 0.5, and 1.0, respectively.

ical modes of the 2-D data, particularly in high-density regions. Notably, $\gamma^2 = 0.1$ achieves the closest visual and structural resemblance to the ground truth, suggesting its suitability for low-dimensional tasks requiring both fidelity and robustness. These results underscore the necessity of tuning $\gamma^2$ to mitigate under-regularization artifacts while preserving distributional complexity.

The ablation study on varying $\gamma^2$ values (Fig. 5) shows distinct convergence behaviors in MI estimation. For all dimensions ($d = \{40, 80, 120\}$), smaller $\gamma^2$ values ($\leq 0.01$) lead to faster convergence, especially in lower dimensions ($d = 40$), but excessively small values ($\gamma^2 = 0.001$) cause instability later. Larger $\gamma^2$ values ($\geq 0.1$) converge more slowly but stabilize over time, particularly in higher dimensions ($d = 120$). $\gamma^2 = 0.1$ offers a balance between speed and stability across all dimensions, suggesting that moderate regularization provides the best MI estimation performance. More results are provided in Appendix C.5.

**Ablation Study on GD.** To evaluate the effectiveness of

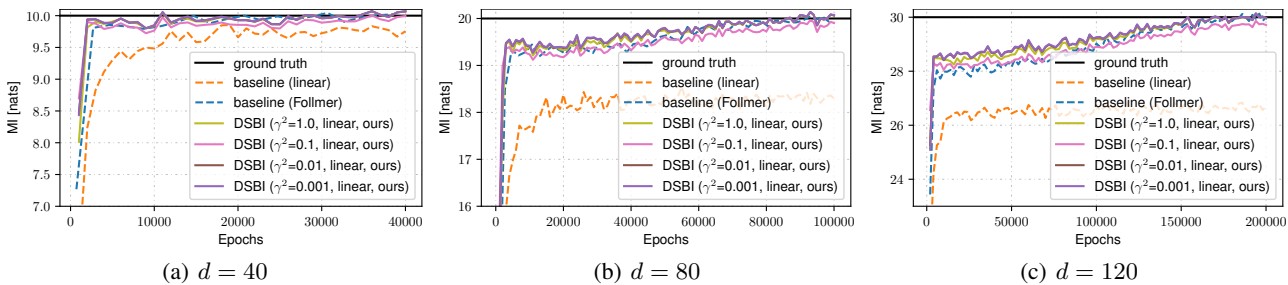

(a) $d = 40$      (b) $d = 80$      (c) $d = 120$

*Figure 5.* Evolution of estimated MI across epochs with varying $\gamma^2 = \{0.001, 0.01, 0.1, 1.0\}$ and dimensions $d = \{40, 80, 120\}$. For all dimensions ($d = \{40, 80, 120\}$), smaller $\gamma^2$ values ($\leq 0.01$) lead to faster convergence.

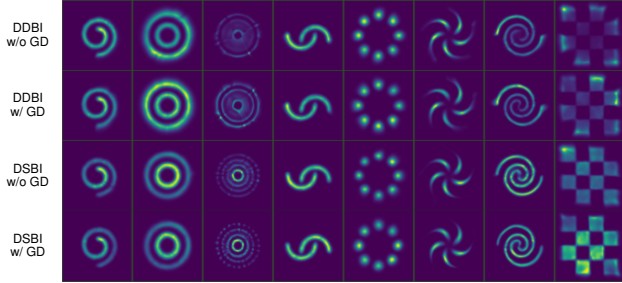

*Figure 6.* Ablation study on GD for eight 2-D synthetic datasets.

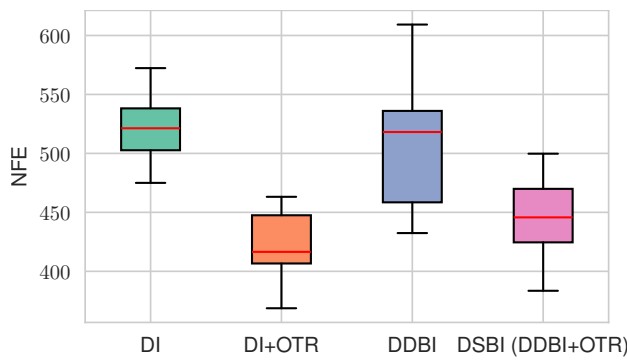

*Figure 7.* Density estimation results on checkerboard for different methods during training (see more results in Figs. 9 to 16).

the proposed GD module, we conduct an ablation study by comparing model performance with and without GD, as shown in Fig. 6. Visually, both DDBI and DSBI show clear improvements in density estimation when GD is applied. Without GD, the estimated densities appear blurrier and miss fine structural details, whereas incorporating GD yields sharper and more realistic patterns.

**Ablation Study on OTR.** We conduct an ablation study to evaluate the role of OTR, comparing models without OTR (baseline, DI), models with OTR (DI+OTR), and models from the $D^3RE$ framework (DDBI and DSBI). In Figs. 2 and 7, DI generates distorted and misaligned intermediate distributions. This shows its limited ability to align with the target distribution. DI+OTR improves alignment but remains suboptimal. Models from the $D^3RE$ further enhance distribution quality, with DSBI achieving the most precise alignment. This underscores OTR's crucial role in improving intermediate distributions.

Fig. 3 compares MI estimation for DDBI and DSBI across dimensions ($d = 80, 120$). Both outperform baseline methods, but DSBI converges faster and remains closer to the ground truth. The advantage of OTR becomes more pronounced in high dimensions ($d = 120$), where DSBI significantly outperforms DDBI in both speed and accuracy.

Overall, OTR improves intermediate distribution alignment. When combined with diffusion bridges and GD, as in DSBI,

it enables more accurate density estimation. Further comparisons are presented in Appendix C.6.

**Number of Function Evaluations.** We analyze the impact of OTR on NFE, noting that DI and DDBI do not utilize OTR. It shows that applying OTR significantly reduces NFE. Fig. 8 compares NFE across four methods in DRE. The first approach exhibits the highest NFE, indicating reliance on iterative procedures requiring repeated function evaluations. The second approach achieves a moderate reduction in NFE, likely by minimizing redundant evaluations through minimized transport costs.

*Figure 8.* Comparison of NFE for four methods in density ratio estimation task. Applying OTR significantly reduces NFE.

# 5. Related Works

**Density Ratio Estimation.** DRE is an essential technique in machine learning but faces challenges in high-dimensional settings and when distributions are significantly different. Early methods (Sugiyama et al., 2012; Gutmann & Hyvärinen, 2012) often struggled with instability, known as the density-chasm problem, in these scenarios. To overcome these challenges, TRE (Rhodes et al., 2020), an extension of NCE (Gutmann & Hyvärinen, 2012), introduced a divide-and-conquer approach, breaking the problem into simpler subproblems for better performance. DRE-∞ (Choi et al., 2022) further advanced this by interpolating between distributions through an infinite series of bridge distributions, improving stability and accuracy. F-DRE (Choi et al., 2021) used an invertible generative model to map distributions into a common feature space before estimation. Recent methods, such as Kato & Teshima (2021), have addressed overfitting in flexible models, and Nagumo & Fujisawa (2024); Luo et al. (2024) focused on improving robustness to outliers. MDRE (Srivastava et al., 2023) tackled distribution shift through multi-class classification, offering an alternative to binary classification in high-discrepancy cases. Additionally, geometric approaches, like Kimura & Bondell (2025), have enhanced DRE accuracy by incorporating the geometry of statistical manifolds. Building on these advancements, our work proposes a novel method to improve both the accuracy and robustness of high-dimensional DRE.

**Diffusion Bridge.** Denoising diffusion implicit models (DDIMs) (Song et al., 2020) have been proposed as an efficient alternative to denoising diffusion probabilistic models (DDPMs) (Ho et al., 2020), which require simulating a Markov chain for many steps to generate samples. Diffusive interpolants (Albergo et al., 2023) provide a unifying framework of flow-based models and diffusion models, bridging arbitrary distributions using continuous-time stochastic processes. DDBMs (Zhou et al., 2024a) are proposed as a natural alternative to cumbersome methods like guidance or projected sampling in generative processes. Our proposed DDBI and DSBI build upon diffusive interpolants and DDBMs by incorporating Brownian bridge into the interpolation strategy construction.

# 6. Conclusions

In this work, we propose $D^3RE$, a unified, robust and efficient framework for density ratio estimation. It provides the first framework for directly addressing both the density-chasm and support-chasm problems, enabling uniformly approximated density-ratio estimation (Proposition 3.5). By incorporating diffusion bridges and GD, we construct DDBI, which expands support coverage (Theorem 3.2, Corollary 3.3) and stabilizes the time score near boundaries (Corollary 3.8). Building upon DDBI, OTR is incorporated

to derive the DSBI, which offers more efficient and stable density ratio estimation (Theorem 3.7) by solving the Schrödinger bridge problem (Proposition 3.6). Together, DDBI and DSBI form the core of the $D^3RE$ framework, enabling uniformly approximated and stable density-ratio estimation (Proposition 3.5). Extensive experiments validate these findings, demonstrating the superior performance of $D^3RE$ in tasks such as density-ratio estimation on synthetic data, mutual information estimation, and density estimation.

While $D^3RE$ advances density ratio estimation methods, several directions remain open. Future work could explore adaptive or learned solvers to reduce function evaluation overhead, as well as more expressive interpolants to further improve robustness in handling complex or multi-modal distributions.

# Acknowledgements

This research work is supported by the Fundamental Research Program of Guangdong, China, under Grant 2023A1515011281.

# Impact Statement

This paper presents work whose goal is to advance the field of Machine Learning. While direct societal impacts are limited, future extensions to applied domains (e.g., via our open-source codebase) should incorporate domain-specific ethical reviews per deployment contexts.

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

## A. Proofs

**Assumption A.1.** Let $q_0, q_1 : \mathbb{R}^d \to \mathbb{R}_+$ be probability density functions satisfying: (1) $q_0, q_1 \in C^2(\mathbb{R}^d)$, i.e., $q_0$ and $q_1$ are twice differentiable and have bounded second derivatives: $\|\nabla_{\mathbf{x}}^2 q_0\|_{L^\infty}, \|\nabla_{\mathbf{x}}^2 q_1\|_{L^\infty} < \infty$; (2) $q_0(\mathbf{x}) > 0$, and there exists $c > 0$ such that $\inf_{\mathbf{x}} q_0(\mathbf{x}) \geq c$. This condition is mild in density ratio estimation.

**Assumption A.2.** The conditional distributions for given $\mathbf{X}_t = \mathbf{x}_t$, i.e., $q_0(\mathbf{x} \mid \mathbf{x}_t)$ and $q_1(\mathbf{x} \mid \mathbf{x}_t)$ have $L$-Lipschitz scores:

$$\|\nabla_{\mathbf{x}_t} \log q_0(\mathbf{x} \mid \mathbf{x}_t)\| \leq \frac{L}{\alpha_t}, \quad \|\nabla_{\mathbf{x}_t} \log q_1(\mathbf{x} \mid \mathbf{x}_t)\| \leq \frac{L}{\beta_t}. \tag{26}$$

### A.1. Proof of Theorem 3.2

*Proof.* We first consider the support for DI. Under $\alpha_t + \beta_t = 1$, the $\mathbf{X}_t$ is the convex combination of $\mathbf{X}_0$ and $\mathbf{X}_1$ for any $t \in (0, 1)$, and its corresponding support, $\mathsf{supp}(q_t)$, is the *convex hull* of $\mathsf{supp}(q_0)$ and $\mathsf{supp}(q_q)$, i.e., $\mathsf{supp}(q_t) = \mathsf{conv}\left(\mathsf{supp}(q_0) \cup \mathsf{supp}(q_1)\right)$. Under $\alpha_t^2 + \beta_t^2 = 1$, $\mathbf{X}_t$ becomes a linear combination of $\mathbf{x}_0$ and $\mathbf{x}_1$ for any $t \in (0, 1)$. For both cases, $\mathsf{supp}(q_t)$ can be formulated as

$$\begin{aligned} \mathsf{supp}(q_t) &= \alpha_t \mathsf{supp}(q_0) + \beta_t \mathsf{supp}(q_1) \\ &= \{\alpha_t \mathbf{x}_0 + \beta_t \mathbf{x}_1 \mid \mathbf{x}_0 \in \mathsf{supp}(q_0), \mathbf{x}_1 \in \mathsf{supp}(q_1), \alpha_t + \beta_t = 1 \text{ or } \alpha_t^2 + \beta_t^2 = 1\}, \end{aligned}$$

where $\mathsf{supp}(q_0)$ and $\mathsf{supp}(q_1)$ are the supports of $q_0$ and $q_1$, respectively.

Next, we consider the support for DBI. For given coefficients $\alpha_t$ and $\beta_t$, $\mathbf{X}_t'$ can be formulated as:

$$\mathbf{X}_t' = \mathbf{I}(\mathbf{X}_0, \mathbf{X}_1, t) + \sqrt{t(1-t)\gamma^2} \mathbf{Z}_t = \mathbf{X}_t + \sqrt{t(1-t)\gamma^2} \mathbf{Z}_t. \tag{27}$$

The coefficient $\sqrt{t(1-t)\gamma^2}$ is deterministic for a given $t$. Thus the support corresponding to $\mathbf{X}_t'$, denoted as $\mathsf{supp}(q_t')$, can be expressed as the Minkowski sum of the supports of $q_t$ and $\mathcal{N}(\mathbf{0}, \mathbf{E}_d)$:

$$\mathsf{supp}(q_t') = \mathsf{supp}(q_t) + \mathsf{supp}(\mathcal{N}(\mathbf{0}, \mathbf{E}_d)) = \{\mathbf{x} + \mathbf{z} \mid \mathbf{x} \in \mathsf{supp}(q_t), z \in \mathsf{supp}(\mathcal{N}(\mathbf{0}, \mathbf{E}_d))\},$$

where $\mathsf{supp}(\mathcal{N}(\mathbf{0}, \mathbf{E}_d)) = \mathbb{R}^d$. The Minkowski sum $\mathsf{supp}(q_t) + \mathsf{supp}(\mathcal{N}(\mathbf{0}, \mathbf{E}_d))$ is at least as large as $\mathsf{supp}(q_t)$, i.e., $\mathsf{supp}(q_t') \supseteq \mathsf{supp}(q_t)$. This completes the proof. $\square$

### A.2. Proof of Corollary 3.3

*Proof.* Let the trajectory sets for DI and DBI be denoted by $\mathbb{T} = \{\{\mathbf{x}_t\}_{t\in[0,1]}; \mathbf{x}_t \in \mathsf{supp}(q_t)\}$ and $\mathbb{T}' = \{\{\mathbf{x}_t'\}_{t\in[0,1]}; \mathbf{x}_t' \in \mathsf{supp}(q_t')\}$, respectively. Let $\{\mathbf{x}_t\}_{t\in[0,1]}$ be an arbitrary element of $\mathbb{T}$. From Theorem 3.2, we have $\mathsf{supp}(q_t') \supseteq \mathsf{supp}(q_t)$ for any $t \in (0, 1)$, meaning that $\mathbf{x}_t \in \mathsf{supp}(q_t')$. Hence we have $\{\mathbf{x}_t\}_{t\in[0,1]} \in \mathbb{T}'$. This directly implies $\mathbb{T}' \supseteq \mathbb{T}$. $\square$

### A.3. Proof of Theorem 3.4

*Proof.* For the DI, the time derivative of its log-density is governed by the Fokker-Planck equation:

$$\partial_t \log q_t = -\nabla \cdot \mathbf{u}_t - \mathbf{u}_t \cdot \nabla \log q_t, \tag{28}$$

where $\nabla \cdot$ and $\nabla$ are the divergence and gradient operators w.r.t. $\mathbf{x}_t$. $\mathbf{u}_t(\mathbf{x}_t) = \mathbb{E}_\pi[\dot{\alpha}_t \mathbf{X}_0 + \dot{\beta}_t \mathbf{X}_1 \mid \mathbf{X}_t = \mathbf{x}_t]$ is the drift term. Taking expectations over $q_t$ and applying the triangle inequality and Cauchy-Schwarz inequality to this equation:

$$\begin{aligned} \mathbb{E}_{q_t}[|\partial_t \log q_t|] &= \mathbb{E}_{q_t}[|-\nabla \cdot \mathbf{u}_t - \mathbf{u}_t \cdot \nabla \log q_t|] \\ &\geq \mathbb{E}_{q_t}[||\nabla \cdot \mathbf{u}_t| - |\mathbf{u}_t \cdot \nabla \log q_t||] \quad \text{(triangle inequality)} \\ &\geq \mathbb{E}_{q_t}[|\nabla \cdot \mathbf{u}_t|] - \mathbb{E}_{q_t}[|\mathbf{u}_t \cdot \nabla \log q_t|] \\ &\geq \mathbb{E}_{q_t}[|\nabla \cdot \mathbf{u}_t|] - \mathbb{E}_{q_t}[\|\mathbf{u}_t\| \cdot \|\nabla \log q_t\|] \quad \text{(Cauchy-Schwarz inequality)} \end{aligned} \tag{29}$$

**(1) Lower bound of $\mathbb{E}_{q_t}[|\nabla \cdot \mathbf{u}_t|]$.** The divergence term $\nabla \cdot \mathbf{u}_t$ is computed via the Jacobian of the inverse mapping:

$$\begin{aligned} \nabla \cdot \mathbf{u}_t = \mathrm{tr}(\nabla \mathbf{u}_t) &= \mathrm{tr}\left(\mathbb{E}_\pi\left[\dot{\alpha}_t \nabla_{\mathbf{x}_t} \mathbf{X}_0 + \dot{\beta}_t \nabla_{\mathbf{x}_t} \mathbf{X}_1 \mid \mathbf{X}_t = \mathbf{x}_t\right]\right) \\ &= \mathbb{E}_\pi\left[\dot{\alpha}_t \mathrm{tr}\left(\nabla_{\mathbf{x}_t} \mathbf{X}_0\right) + \dot{\beta}_t \mathrm{tr}\left(\nabla_{\mathbf{x}_t} \mathbf{X}_1\right) \mid \mathbf{x}_t\right]. \end{aligned} \tag{30}$$

For a given sample $(\mathbf{x}_0, \mathbf{x}_1) \sim \pi$, differentiating the the interpolation constraint $\mathbf{x}_t = \alpha_t \mathbf{x}_0 + \beta_t \mathbf{x}_1$ implicitly gives $\mathbf{E}_d = \alpha_t \nabla_{\mathbf{x}_t} \mathbf{x}_0 + \beta_t \nabla_{\mathbf{x}_t} \mathbf{x}_1$. Rearranging yields $\nabla_{\mathbf{x}_t} \mathbf{x}_0 = \alpha_t^{-1} \mathbf{E}_d - \beta_t \alpha_t^{-1} \nabla_{\mathbf{x}_t} \mathbf{x}_1$ and $\nabla_{\mathbf{x}_t} \mathbf{x}_1 = \beta_t^{-1} \mathbf{E}_d - \alpha_t \beta_t^{-1} \nabla_{\mathbf{x}_t} \mathbf{x}_0$.

Decomposing $\nabla_{\mathbf{x}_t} \mathbf{x}_0$ into its independent case value and a residual:

$$\nabla_{\mathbf{x}_t} \mathbf{x}_0 = \alpha_t^{-1} \mathbf{E}_d + \mathbf{R}_t, \tag{31}$$

where $\mathbf{R}_t = -\beta_t \alpha_t^{-1} \nabla_{\mathbf{x}_t} \mathbf{x}_1$ denote the residual term in the Jacobian decomposition. By Assumption A.2 and the definition of the score function of the conditional distribution, $\nabla_{\mathbf{x}_t} \log q_0(\mathbf{x}_0 \mid \mathbf{x}_t) = -\nabla_{\mathbf{x}_t} \mathbf{x}_0 \cdot \nabla_{\mathbf{x}_0} \log q_0(\mathbf{x}_0 \mid \mathbf{x}_t)$, the Lipschitz continuity of the conditional scores implies $\|\nabla_{\mathbf{x}_t} \mathbf{x}_0\|_{\mathrm{op}} \leq L \alpha_t^{-1}$ and $\|\nabla_{\mathbf{x}_t} \mathbf{x}_1\|_{\mathrm{op}} \leq L \beta_t^{-1}$. Here $\|\cdot\|_{\mathrm{op}}$ denotes the operator norm. Hence, the absolute value of the trace of the residual term satisfies:

$$|\mathrm{tr}(\mathbf{R}_t)| \leq d \cdot \|\mathbf{R}_t\|_{\mathrm{op}} = d\alpha_t^{-1} \beta_t \|\nabla_{\mathbf{x}_t} \mathbf{X}_1\|_{\mathrm{op}} \leq d\alpha_t^{-1} \beta_t \cdot \frac{L}{\beta_t} = dL\alpha_t^{-1}. \tag{32}$$

This directly leads to the conditional expectation: $|\mathbb{E}_{q_t}[\mathrm{tr}(\mathbf{R}_t) \mid \mathbf{X}_t]| \leq \mathbb{E}_{q_t}[|\mathrm{tr}(\mathbf{R}_t)| \mid \mathbf{X}_t] \leq dL\alpha_t^{-1}$. Thus, the conditional expectation of $\mathrm{tr}(\nabla_{\mathbf{x}_t} \mathbf{X}_0)$ satisfies:

$$
\begin{aligned}
|\mathbb{E}_{q_t}[\mathrm{tr}(\nabla_{\mathbf{x}_t} \mathbf{X}_0) \mid \mathbf{X}_t]| &= \left| \mathbb{E}_{q_t}\left[\mathrm{tr}\left(\alpha_t^{-1} \mathbf{E}_d + \mathbf{R}_t\right) \mid \mathbf{X}_t\right] \right| = \left| \mathbb{E}_{q_t}\left[d\alpha_t^{-1} + \mathrm{tr}(\mathbf{R}_t) \mid \mathbf{X}_t\right] \right| \\
&= \left| d\alpha_t^{-1} + \mathbb{E}_{q_t}[\mathrm{tr}(\mathbf{R}_t) \mid \mathbf{X}_t] \right| \\
&\geq d\alpha_t^{-1} - |\mathbb{E}_{q_t}[\mathrm{tr}(\mathbf{R}_t) \mid \mathbf{X}_t]| \\
&\geq d\alpha_t^{-1} - dL\alpha_t^{-1} = d\alpha_t^{-1}(1 - L).
\end{aligned}
\tag{33}
$$

Similarly, we have $|\mathbb{E}_{q_t}[\mathrm{tr}(\nabla_{\mathbf{x}_t} \mathbf{X}_1) \mid \mathbf{x}_t]| \geq d\beta_t^{-1}(1 - L)$. Base on these lower bounds and applying the triangle inequality, we have:

$$
\begin{aligned}
\mathbb{E}_{q_t}[|\nabla \cdot \mathbf{u}_t|] &= \mathbb{E}_{q_t}\left[ \left| \mathbb{E}_{q_t}\left[ \dot{\alpha}_t \mathrm{tr}(\nabla_{\mathbf{x}_t} \mathbf{X}_0) + \dot{\beta}_t \mathrm{tr}(\nabla_{\mathbf{x}_t} \mathbf{X}_1) \mid \mathbf{X}_t \right] \right| \right] \\
&\geq \mathbb{E}_{q_t}\left[ |\dot{\alpha}_t| |\mathbb{E}_{q_t}[\mathrm{tr}(\nabla_{\mathbf{x}_t} \mathbf{X}_0) \mid \mathbf{X}_t]| - |\dot{\beta}_t| |\mathbb{E}_{q_t}[\mathrm{tr}(\nabla_{\mathbf{x}_t} \mathbf{X}_1) \mid \mathbf{X}_t]| \right] \\
&\geq \mathbb{E}_{q_t}\left[ d|\dot{\alpha}_t| \alpha_t^{-1}(1 - L) - |\dot{\beta}_t| \mathbb{E}_{q_t}[|\mathrm{tr}(\nabla_{\mathbf{x}_t} \mathbf{X}_1)| \mid \mathbf{X}_t] \right] \\
&\geq \mathbb{E}_{q_t}\left[ d|\dot{\alpha}_t| \alpha_t^{-1}(1 - L) - |\dot{\beta}_t| \cdot dL\beta_t^{-1} \right] \quad \left( |\mathrm{tr}(\nabla_{\mathbf{x}_t} \mathbf{X}_1)| \leq d \cdot \|\nabla_{\mathbf{x}_t} \mathbf{X}_1\|_{\mathrm{op}} \leq dL\beta_t^{-1} \right) \\
&= d\left( (1 - L)|\dot{\alpha}_t| \alpha_t^{-1} - L|\dot{\beta}_t| \beta_t^{-1} \right).
\end{aligned}
\tag{34}
$$

**(2) Upper bound of $\mathbb{E}_{q_t}[\|\mathbf{u}_t\| \cdot \|\nabla \log q_t\|]$.** Applying Cauchy-Schwarz:

$$
\begin{aligned}
\mathbb{E}_{q_t}[\|\mathbf{u}_t\| \cdot \|\nabla \log q_t\|] &\leq \sqrt{\mathbb{E}_{q_t}[\|\mathbf{u}_t\|^2]} \cdot \sqrt{\mathbb{E}_{q_t}[\|\nabla \log q_t\|^2]} \\
&= \sqrt{\mathbb{E}_{q_t}\left[ \left\| \mathbb{E}_{\pi}\left[ \dot{\alpha}_t \mathbf{X}_0 + \dot{\beta}_t \mathbf{X}_1 | \mathbf{X}_t \right] \right\|^2 \right]} \cdot \sqrt{\mathbb{E}_{q_t}[\|\nabla \log q_t\|^2]} \\
&\leq \sqrt{\mathbb{E}_{q_t}\left[ \mathbb{E}_{\pi}\left[ \|\dot{\alpha}_t \mathbf{X}_0 + \dot{\beta}_t \mathbf{X}_1\|^2 | \mathbf{X}_t \right] \right]} \cdot \sqrt{\mathbb{E}_{q_t}[\|\nabla \log q_t\|^2]} \\
&\leq \sqrt{2\dot{\alpha}_t^2 \mathbb{E}_{q_0}[\|\mathbf{X}_0\|^2] + 2\dot{\beta}_t^2 \mathbb{E}_{q_1}[\|\mathbf{X}_1\|^2]} \cdot \sqrt{\mathbb{E}_{q_t}[\|\nabla \log q_t\|^2]} = \mathcal{O}\left( \sqrt{\mathbb{E}_{q_t}[\|\nabla \log q_t\|^2]} \right).
\end{aligned}
\tag{35}
$$

Here the drift norm $\mathbb{E}_{q_t}[\|\mathbf{u}_t\|^2]$ is bounded by the second moments of $\mathbf{X}_0$ and $\mathbf{X}_1$, which are finite under our assumptions.

**(3) Lower bound of $\mathbb{E}_{q_t}[|\partial_t \log q_t|]$.** Finally, bring Eq. (34) and Eq. (35) back to Eq. (29), the lower bound of $\mathbb{E}_{q_t}[|\partial_t \log q_t|]$ can be derived:

$$
\begin{aligned}
\mathbb{E}_{q_t}[|\partial_t \log q_t|] &\geq \mathbb{E}_{q_t}[|\nabla \cdot \mathbf{u}_t|] - \mathbb{E}_{q_t}[\|\mathbf{u}_t\| \cdot \|\nabla \log q_t\|] \\
&\geq \underbrace{d\left( (1 - L)\frac{|\dot{\alpha}_t|}{\alpha_t} - L\frac{|\dot{\beta}_t|}{\beta_t} \right)}_{\text{Divergence term}} - \underbrace{\mathcal{O}\left( \sqrt{\mathbb{E}_{q_t}[\|\nabla \log q_t\|^2]} \right)}_{\text{Residual term}}.
\end{aligned}
\tag{36}
$$

The residual term $\mathcal{O}(\sqrt{\mathbb{E}_{q_t}[\|\nabla \log q_t\|^2]})$ is finite under Assumption A.2 (Lipschitz scores imply bounded Fisher information). Hence, the divergence term dominates asymptotically.

Near the boundaries $t \to 0^+$ and $t \to 1^-$, the terms $\frac{|\dot{\alpha}_t|}{\alpha_t}$ and $\frac{|\dot{\beta}_t|}{\beta_t}$ dominate due to the monotonicity and boundary conditions:

$$(1) \text{ As } t \to 0^+ : \alpha_t \to 1, \quad \beta_t \to 0, \quad \frac{|\dot{\alpha}_t|}{\alpha_t} \sim |\dot{\alpha}_0|, \quad \frac{|\dot{\beta}_t|}{\beta_t} \sim \frac{\dot{\beta}_0^+}{\beta_t} \to +\infty.$$

$$(2) \text{ As } t \to 1^- : \alpha_t \to 0, \quad \beta_t \to 1, \quad \frac{|\dot{\alpha}_t|}{\alpha_t} \sim \frac{|\dot{\alpha}_1^-|}{\alpha_t} \to +\infty, \quad \frac{|\dot{\beta}_t|}{\beta_t} \sim |\dot{\beta}_1|.$$

(37)

For any $L < 1$, the prefactor $1 - L > 0$ ensures:

$$\lim_{t \to 1^-} \mathbb{E}_{q_t}[\|\partial_t \log q_t\|] \geq \lim_{t \to 1^-} d(1 - L)\frac{|\dot{\alpha}_t|}{\alpha_t} = +\infty. \tag{38}$$

This concludes the universal boundary divergence.

$\square$

## A.4. Proof of Proposition 3.5

*Proof.* Assume both $q_0$ and $q_1$ are smooth with sufficient differentiability. The Gaussian-dequantified densities are given by:

$$
\begin{aligned}
q_i'(\mathbf{x}) &= (q_i * \mathcal{N}(\mathbf{0}, \varepsilon \mathbf{E}_d))(\mathbf{x}) = \int q_i(\mathbf{x}')\mathcal{N}(\mathbf{x}; \mathbf{x}', \varepsilon \mathbf{E}_d)d\mathbf{x}' \\
&= \int q_i(\mathbf{x}')\left[\delta(\mathbf{x} - \mathbf{x}') + \frac{\varepsilon}{2}\nabla_{\mathbf{x}'}^2\delta(\mathbf{x} - \mathbf{x}') + \mathcal{O}(\varepsilon^2)\right]d\mathbf{x}' \quad \text{(Taylor expansion around } \mathbf{x}') \\
&= q_i(\mathbf{x}) + \frac{\varepsilon}{2}\int q_i(\mathbf{x}')\nabla_{\mathbf{x}'}^2\delta(\mathbf{x} - \mathbf{x}')d\mathbf{x}' + \mathcal{O}(\varepsilon^2) \\
&= q_i(\mathbf{x}) + \frac{\varepsilon}{2}\nabla_{\mathbf{x}}^2 q_i(\mathbf{x}) + \mathcal{O}(\varepsilon^2), \quad \text{(Integration by parts)},
\end{aligned}
$$

(39)

where $\nabla_{\mathbf{x}'}^2$ is the Laplacian operator, $\delta(\mathbf{x} - \mathbf{x}')$ is the Dirac delta.

Substituting these two expansions into the dequantified density ratio $r'(\mathbf{x}) = \frac{q_1'(\mathbf{x})}{q_0'(\mathbf{x})}$, we have:

$$
\begin{aligned}
r'(\mathbf{x}) &= \frac{q_1'(\mathbf{x})}{q_0'(\mathbf{x})} = \frac{q_1(\mathbf{x}) + \frac{\varepsilon}{2}\nabla_{\mathbf{x}}^2 q_1(\mathbf{x}) + \mathcal{O}(\varepsilon^2)}{q_0(\mathbf{x}) + \frac{\varepsilon}{2}\nabla_{\mathbf{x}}^2 q_0(\mathbf{x}) + \mathcal{O}(\varepsilon^2)} \\
&= \frac{q_1(\mathbf{x})}{q_0(\mathbf{x})} + \frac{\varepsilon}{2}\frac{\nabla_{\mathbf{x}}^2 q_1(\mathbf{x})}{q_0(\mathbf{x})} - \frac{\varepsilon}{2}r(\mathbf{x})\frac{\nabla_{\mathbf{x}}^2 q_0(\mathbf{x})}{q_0(\mathbf{x})} + \mathcal{O}(\varepsilon^2) \quad \text{(First-order expansion of a fraction)} \\
&= r(\mathbf{x}) + \frac{\varepsilon}{2}\underbrace{\left[\frac{\nabla_{\mathbf{x}}^2 q_1(\mathbf{x})}{q_0(\mathbf{x})} - r(\mathbf{x})\frac{\nabla_{\mathbf{x}}^2 q_0(\mathbf{x})}{q_0(\mathbf{x})}\right]}_{\Delta(\mathbf{x})} + \mathcal{O}(\varepsilon^2).
\end{aligned}
$$

(40)

To bound $r'(\mathbf{x}) - r(\mathbf{x})$ in $L^\infty$, the supremum can be computed under Assumption A.1:

$$
\begin{aligned}
\|r' - r\|_{L^\infty} &\leq \frac{\varepsilon}{2}\sup_{\mathbf{x}}|\Delta(\mathbf{x})| + \mathcal{O}(\varepsilon^2) \\
&= \frac{\varepsilon}{2}\sup_{\mathbf{x}}\left|\frac{\nabla_{\mathbf{x}}^2 q_1(\mathbf{x})}{q_0(\mathbf{x})} - r(\mathbf{x})\frac{\nabla_{\mathbf{x}}^2 q_0(\mathbf{x})}{q_0(\mathbf{x})}\right| + \mathcal{O}(\varepsilon^2) \\
&\leq \frac{\varepsilon}{2}\sup_{\mathbf{x}}\left|\frac{\nabla_{\mathbf{x}}^2 q_1(\mathbf{x})}{q_0(\mathbf{x})}\right| + \frac{\varepsilon}{2}\sup_{\mathbf{x}}\left|r(\mathbf{x})\frac{\nabla_{\mathbf{x}}^2 q_0(\mathbf{x})}{q_0(\mathbf{x})}\right| + \mathcal{O}(\varepsilon^2) \\
&\leq \frac{\varepsilon}{2}\left(\frac{\|\nabla_{\mathbf{x}}^2 q_1\|_{L^\infty}}{\inf_{\mathbf{x}} q_0(\mathbf{x})} + \frac{\|\nabla_{\mathbf{x}}^2 q_0\|_{L^\infty}}{\inf_{\mathbf{x}} q_0(\mathbf{x})}\sup_{\mathbf{x}} r(\mathbf{x})\right) + \mathcal{O}(\varepsilon^2) \\
&= \frac{\varepsilon}{2\inf_{\mathbf{x}} q_0(\mathbf{x})}\left(\|\nabla_{\mathbf{x}}^2 q_1\|_{L^\infty} + \|\nabla_{\mathbf{x}}^2 q_0\|_{L^\infty}\|r\|_{L^\infty}\right) + \mathcal{O}(\varepsilon^2).
\end{aligned}
$$

(41)

Under Assumption A.1, $\inf_{\mathbf{x}} q_0(\mathbf{x})$ has lower bound and these norms have upper bound. Then as $\varepsilon \to 0$,

$$\|r' - r\|_{L^\infty} \leq \mathcal{O}(\varepsilon), \tag{42}$$

where the constant $C$ is given by: $C = \frac{\|\nabla_{\mathbf{x}}^2 q_1\|_{L^\infty} + \|\nabla_{\mathbf{x}}^2 q_0\|_{L^\infty} \|r\|_{L^\infty}}{2 \inf_{\mathbf{x}} q_0(\mathbf{x})}$. Hence, as $\varepsilon \to 0$, we have $r'(\mathbf{x}) \to r(\mathbf{x})$, verifying the stated proposition. $\square$

### A.5. Proof of Proposition 3.6

*Proof.* Based on Theorem 2.4 of (Léonard, 2014), (De Bortoli et al., 2021) established that the solution to the SB problem as detailed in Eq. (19), $\pi^\star$, satisfies the SB conditions: (1) the optimization problem for $\pi^\star$ is equivalent to the entropically regularied OT problem, with the optimal coupling $\pi_{2\gamma^2}$ defined in Eq. (17); (2) for samples $(\mathbf{x}_0', \mathbf{x}_1') \sim \pi^\star$, the associated conditional path distributions $\pi^\star(\cdot \mid \mathbf{x}_0', \mathbf{x}_1')$ minimize the KL divergence: $\mathbb{E}_{(\mathbf{x}_0', \mathbf{x}_1') \sim \pi^\star} \mathrm{KL}\left(\pi^\star(\cdot \mid \mathbf{x}_0', \mathbf{x}_1') \| \pi_{\mathrm{ref}}(\cdot \mid \mathbf{x}_0', \mathbf{x}_1')\right)$, where $\pi_{\mathrm{ref}}$ is the reference path distribution satisfying $\log \pi_{\mathrm{ref}}(\mathbf{x}_0', \mathbf{x}_1') = \frac{\|\mathbf{x}_0' - \mathbf{x}_1'\|^2}{2\gamma^2} + \mathrm{const}$. These conditional distributions are optimized using Brownian bridges of diffusion scale $\gamma$, conditioned on the endpoints $\mathbf{x}_0'$ and $\mathbf{x}_1'$. The marginal distribution at intermediate time $t$ along the Brownian bridge is given by $q_t(\mathbf{x} \mid \mathbf{x}_0', \mathbf{x}_1') = \mathcal{N}(\mathbf{x} \mid (1-t)\mathbf{x}_0' + t\mathbf{x}_1', t(1-t)\gamma^2 \mathbf{E}_d)$ (Tong et al., 2024). Since our proposed OTR method uses Sinkhorn's algorithm to solve the entropically regularized OT problem, and the probability paths of our DDBI with $\alpha_t = 1 - t$ and $\beta_t = t$ align with those of the Brownian bridge, a trajectory $\mathbf{x}_t'$ generated by first sampling $(\mathbf{x}_0', \mathbf{x}_1') \sim \pi^\star$, then sampling $\mathbf{x}_t \sim q_t(\cdot \mid \mathbf{x}_0', \mathbf{x}_1')$ satisfies the SB conditions, thus verifying the proposition. $\square$

### A.6. Proof of Theorem 3.7

*Proof.* To analyze $\mathrm{Var}_{q_t'}(\partial_t \log q_t')$, applying the law of total variance to $\mathrm{Var}_{q_t'}(\partial_t \log q_t')$:

$$\mathrm{Var}_{q_t'}(\partial_t \log q_t') = \mathbb{E}_{(\mathbf{X}_0', \mathbf{X}_1') \sim \pi} \left[ \mathrm{Var}\left(\partial_t \log q_t'(\mathbf{X}_t' \mid \mathbf{X}_0', \mathbf{X}_1')\right)\right] + \mathrm{Var}_{(\mathbf{X}_0', \mathbf{X}_1') \sim \pi}\left(\mathbb{E}\left[\partial_t \log q_t'(\mathbf{X}_t' \mid \mathbf{X}_0', \mathbf{X}_1')\right]\right). \tag{43}$$

For any paired endpoints $(\mathbf{X}_0', \mathbf{X}_1') \sim \pi$, the interpolant $\mathbf{X}_t'$ follows a Gaussian distribution conditioned on the endpoints $\mathbf{X}_t' \mid (\mathbf{X}_0', \mathbf{X}_1') \sim q_t'(\cdot \mid \mathbf{X}_0', \mathbf{X}_1') = \mathcal{N}\left(\boldsymbol{\mu}_t, \sigma_t^2 \mathbf{I}_d\right)$, where $\boldsymbol{\mu}_t = \alpha_t \mathbf{X}_0' + \beta_t \mathbf{X}_1'$, $\sigma_t^2 = t(1-t)\gamma^2 + (\alpha_t^2 + \beta_t^2)\varepsilon$. The conditional score $\partial_t \log q_t'(\mathbf{X}_t' \mid \mathbf{X}_0', \mathbf{X}_1')$ is derived explicitly as:

$$\begin{aligned}
\partial_t \log q_t'(\mathbf{X}_t' \mid \mathbf{X}_0', \mathbf{X}_1') &= \frac{\partial}{\partial t} \log \mathcal{N}\left(\boldsymbol{\mu}_t, \sigma_t^2 \mathbf{I}_d\right) = \frac{\partial}{\partial t} \log \left[ (2\pi\sigma_t^2)^{-d/2} \exp\left(-\frac{\|\mathbf{X}_t' - \boldsymbol{\mu}_t\|^2}{2\sigma_t^2}\right)\right] \\
&= -\frac{d}{2}\frac{\dot{\sigma}_t^2}{\sigma_t^2} + \frac{\dot{\alpha}_t \mathbf{X}_0' + \dot{\beta}_t \mathbf{X}_1'}{\sigma_t^2} \cdot (\mathbf{X}_t' - \boldsymbol{\mu}_t) + \frac{\|\mathbf{X}_t' - \boldsymbol{\mu}_t\|^2}{2}\frac{\dot{\sigma}_t^2}{\sigma_t^4} \\
&= -\frac{d\dot{\sigma}_t^2}{2\sigma_t^2} + \frac{\dot{\alpha}_t \mathbf{X}_0' + \dot{\beta}_t \mathbf{X}_1'}{\sigma_t} \cdot \mathbf{Z}_t + \frac{\dot{\sigma}_t^2 \|\mathbf{Z}_t\|^2}{2\sigma_t^2}.
\end{aligned} \tag{44}$$

**The term** $\mathbb{E}_\pi \left[\mathrm{Var}\left(\partial_t \log q_t'(\mathbf{X}_t' \mid \mathbf{X}_0', \mathbf{X}_1')\right)\right]$**.** Taking the variance on both sides of Eq. (44) and taking expectation over the coupling distribution $\pi$:

$$\begin{aligned}
\mathbb{E}_\pi \left[\mathrm{Var}(\partial_t \log q_t'(\mathbf{X}_t' \mid \mathbf{X}_0', \mathbf{X}_1'))\right] &= \mathbb{E}_\pi \left[ \mathrm{Var}\left(\frac{\dot{\alpha}_t \mathbf{X}_0' + \dot{\beta}_t \mathbf{X}_1'}{\sigma_t} \cdot \mathbf{Z}_t\right) + \mathrm{Var}\left(-\frac{d\dot{\sigma}_t^2}{2\sigma_t^2}\right) + \mathrm{Var}\left(-\frac{\dot{\sigma}_t^2 \|\mathbf{Z}_t\|^2}{2\sigma_t^2}\right)\right] \\
&= \frac{1}{\sigma_t^2}\mathbb{E}_\pi \left[\left\|\dot{\alpha}_t \mathbf{X}_0' + \dot{\beta}_t \mathbf{X}_1'\right\|^2 + 0 + \frac{\dot{\sigma}_t^4}{4\sigma_t^4}\mathrm{Var}(\|\mathbf{Z}_t\|^2)\right] \\
&= \frac{1}{\sigma_t^2}\mathbb{E}_\pi \left[\left\|\dot{\alpha}_t \mathbf{X}_0' + \dot{\beta}_t \mathbf{X}_1'\right\|^2\right] + \frac{\dot{\sigma}_t^4 d}{2\sigma_t^4}.
\end{aligned} \tag{45}$$

Specifically, when $\alpha_t = 1 - t$ and $\beta_t = t$, this reduces to $\mathbb{E}_\pi \left[\mathrm{Var}(\partial_t \log q_t'(\mathbf{X}_t' \mid \mathbf{X}_0', \mathbf{X}_1'))\right] = \frac{1}{\sigma_t^2}\mathbb{E}_\pi \left[\|\mathbf{X}_0' - \mathbf{X}_1'\|^2\right] + \frac{\dot{\sigma}_t^4 d}{2\sigma_t^4}$.

**The term** $\mathrm{Var}\left(\mathbb{E}\left[\partial_t \log q_t'(\mathbf{X}_t' \mid \mathbf{X}_0', \mathbf{X}_1')\right]\right)$**.** Taking the variance on both sides of Eq. (44) and taking expectation over the

coupling distribution $\pi$:

$$
\begin{aligned}
\mathrm{Var}_\pi \left( \mathbb{E} \left[ \partial_t \log q_t'(\mathbf{X}_t' \mid \mathbf{X}_0', \mathbf{X}_1') \right] \right) &= \mathrm{Var}_\pi \left( \mathbb{E} \left[ -\frac{d\dot\sigma_t^2}{2\sigma_t^2} + \frac{\dot\alpha_t \mathbf{X}_0' + \dot\beta_t \mathbf{X}_1'}{\sigma_t} \cdot \mathbf{Z}_t + \frac{\dot\sigma_t^2 \|\mathbf{Z}_t\|^2}{2\sigma_t^2} \right] \right) \\
&= \mathrm{Var}_\pi \left( -\frac{d\dot\sigma_t^2}{2\sigma_t^2} + \frac{\dot\alpha_t \mathbf{X}_0' + \dot\beta_t \mathbf{X}_1'}{\sigma_t} \cdot \mathbb{E}[\mathbf{Z}_t] + \frac{\dot\sigma_t^2}{2\sigma_t^2} \mathbb{E}\left[ \|\mathbf{Z}_t\|^2 \right] \right) \\
&= \mathrm{Var}_\pi \left( \frac{\dot\alpha_t \mathbf{X}_0' + \dot\beta_t \mathbf{X}_1'}{\sigma_t} \cdot \mathbf{0} + \frac{\dot\sigma_t^2}{2\sigma_t^2} d \right) = 0.
\end{aligned}
\tag{46}
$$

The last equality holds because $\mathbf{Z}_t$ is a Gaussian noise, leading to $\mathbf{Z}_t \sim \mathcal{N}(\mathbf{0}, \mathbf{E}_d)$ and $\|\mathbf{Z}_t\|^2 \sim \chi(d)$.

Bringing Eqs. (45) and (46) into Eq. (43), the variance term $\mathrm{Var}_{q_t'}(\partial_t \log q_t')$ becomes:

$$
\mathrm{Var}_{q_t'}(\partial_t \log q_t') = \int_0^1 \left[ \frac{1}{\sigma_t^2} \mathbb{E}_\pi \left[ \|\mathbf{X}_0' - \mathbf{X}_1'\|^2 \right] + \frac{\dot\sigma_t^4 d}{2\sigma_t^4} \right] \mathrm{d}t = \mathbb{E}_\pi \left[ \|\mathbf{X}_0' - \mathbf{X}_1'\|^2 \right] \int_0^1 \frac{1}{\sigma_t^2} \mathrm{d}t + \int_0^1 \frac{\dot\sigma_t^4 d}{2\sigma_t^4} \mathrm{d}t, \tag{47}
$$

when $\alpha_t = 1 - t$ and $\beta_t = t$. Thus, the difference between the upper bound of the variance for DSBI and DDBI becomes:

$$
\begin{aligned}
&\mathrm{Var}_{q_t'}^{\mathrm{DDBI}}(\partial_t \log q_t') - \mathrm{Var}_{q_t'}^{\mathrm{DSBI}}(\partial_t \log q_t') \\
=&\mathbb{E}_\pi \left[ \|\mathbf{X}_0' - \mathbf{X}_1'\|^2 \right] - \mathbb{E}_{\pi_{2\gamma^2}} \left[ \left\| \hat{\mathbf{X}}_0' - \hat{\mathbf{X}}_1' \right\|^2 \right] \\
=& \left[ \mathbb{E}_\pi \left[ \|\mathbf{X}_0' - \mathbf{X}_1'\|^2 \right] - 2\gamma^2 \mathcal{H}(\pi_{2\gamma^2}) \right] - \left[ \mathbb{E}_{\pi_{2\gamma^2}} \left[ \left\| \hat{\mathbf{X}}_0' - \hat{\mathbf{X}}_1' \right\|^2 \right] - 2\gamma^2 \mathcal{H}(\pi_{2\gamma^2}) \right] \geq 0.
\end{aligned}
\tag{48}
$$

This inequality holds because $\pi_{2\gamma^2}$ is the solution to the entropically regularized OT problem. This completes the proof.

$\square$

## A.7. Proof of Corollary 3.8

*Proof.* Directly applying the Jensen's inequality to $\mathbb{E}_{q_t'}[|\partial_t \log q_t'|]$ yields:

$$
\mathbb{E}_{q_t'}[|\partial_t \log q_t'|] \leq \sqrt{\mathbb{E}_{q_t'}[(\partial_t \log q_t')^2]} = \sqrt{\mathrm{Var}_{q_t'}(\partial_t \log q_t') + \left( \mathbb{E}_{q_t'}[\partial_t \log q_t'] \right)^2}. \quad \text{(Property of variance).} \tag{49}
$$

The first term on the r.h.s. is bounded according to Theorem 3.7, satisfying $\mathrm{Var}_{q_t'}(\partial_t \log q_t') = \frac{1}{\sigma_t^2} \mathbb{E}_\pi \left[ \|\mathbf{X}_0' - \mathbf{X}_1'\|^2 \right] + \frac{\dot\sigma_t^4 d}{2\sigma_t^4}$ with $\sigma_t^2 = t(1-t)\gamma^2 + (\alpha_t^2 + \beta_t^2)\varepsilon$. The second term on the r.h.s. vanishes identically $\mathbb{E}_{q_t'}[\partial_t \log q_t'] = \int \frac{\partial_t q_t'}{q_t'} q_t' \mathrm{d}\mathbf{x} = \int \partial_t q_t' \mathrm{d}\mathbf{x} = \partial_t \left( \int q_t' \mathrm{d}\mathbf{x} \right) = \partial_t(1) = 0$. Thus, the term $\mathbb{E}_{q_t'}[|\partial_t \log q_t'|]$ is bounded by:

$$
\mathbb{E}_{q_t'}[|\partial_t \log q_t'|] \leq \sqrt{\frac{1}{\sigma_t^2} \mathbb{E}_\pi \left[ \|\mathbf{X}_0' - \mathbf{X}_1'\|^2 \right] + \frac{\dot\sigma_t^4 d}{2\sigma_t^4}} < \infty. \tag{50}
$$

$\square$

## A.8. Derivation of Definition 3.9

*Proof.* Let $\{\mathbf{X}_t'\}_{t \in [0,1]}$ be a DDBI. It has a transition kernel $q_t'(\mathbf{x} \mid \mathbf{x}_0, \mathbf{x}_1)$ and marginal probability density $q_t'(\mathbf{x})$. By dividing the interval $[0, 1]$ into $M$ discrete intervals, the log dequantified density ratio for a given point $\mathbf{x}$ can be derived:

$$
\log r'(\mathbf{x}) = \log \frac{q_1'(\mathbf{x})}{q_0'(\mathbf{x})} = \log \frac{q_{1/M}'(\mathbf{x})}{q_0'(\mathbf{x})} \frac{q_{2/M}'(\mathbf{x})}{q_{1/M}'(\mathbf{x})} \cdots \frac{q_1'(\mathbf{x})}{q_{(M-1)/M}'(\mathbf{x})} = \sum_{m=0}^{M-1} \log \frac{q_{(m+1)/M}'(\mathbf{x})}{q_{m/M}'(\mathbf{x})}. \tag{51}
$$

According to the Taylor's formula, we have $\log(1 + \mathbf{x}) \approx \mathbf{x}$ while $\mathbf{x}$ approaches 0. In this case, while $M$ is large enough so that the difference between $p_{m/M}(\mathbf{x} \mid \mathbf{x}_0, \mathbf{x}_1)$ and $p_{(m-1)/M}(\mathbf{x} \mid \mathbf{x}_0, \mathbf{x}_1)$ approaches 0, we have

$$\log \frac{q'_{(m+1)/M}(\mathbf{x})}{q'_{m/M}(\mathbf{x})} = \log\left(1 + \frac{q'_{(m+1)/M}(\mathbf{x}) - q'_{m/M}(\mathbf{x})}{q'_{m/M}(\mathbf{x})}\right) \approx \frac{q'_{(m+1)/M}(\mathbf{x}) - q'_{m/M}(\mathbf{x})}{q'_M(\mathbf{x})}. \tag{52}$$

In the limit as $M \to \infty$, the difference term $\frac{q'_{(m+1)/M}(\mathbf{x}) - q'_{m/M}(\mathbf{x})}{q'_M(\mathbf{x})}$ can be seen as the approximation of $\frac{\partial}{\partial \tau} \log q'_\tau(\mathbf{x})$ evaluated at $\tau = m/M$. Taking the limit as $M \to \infty$ for both sides of Eq. (51), we can derive

$$\begin{aligned}
\log r'(\mathbf{x}) &= \lim_{M \to \infty} \sum_{m=0}^{M-1} \log \frac{q'_{(m+1)/M}(\mathbf{x})}{q'_{m/M}(\mathbf{x})} \\
&\approx \lim_{M \to \infty} \sum_{m=0}^{M-1} \frac{q'_{(m+1)/M}(\mathbf{x}) - q'_{m/M}(\mathbf{x})}{q'_M(\mathbf{x})} \\
&\approx \lim_{M \to \infty} \sum_{m=0}^{M-1} \left.\frac{\partial}{\partial \tau} \log q'_\tau(\mathbf{x})\right|_{\tau = m/M} \\
&= \int_0^1 \partial_t \log q'_\tau(\mathbf{x}) \mathrm{d}t.
\end{aligned} \tag{53}$$

According to Proposition 3.5, the density ratio $r'(\mathbf{x})$ uniformly approximates the target density ratio $r^\star(\mathbf{x})$, i.e.

$$\log r(\mathbf{x}) \approx \log r'(\mathbf{x}) \approx \int_0^1 \partial_t \log q'_\tau(\mathbf{x}) \mathrm{d}t. \tag{54}$$

This completes the proof of this proposition. $\qquad\square$

## B. Preliminaries

### B.1. Special Cases of DI

In the case of TRE, the interpolation strategy, as detailed in Eq. (2), represents a specific case of $\mathbf{I}$, characterized by $\alpha_t = \sqrt{1 - \eta_t^2}, \beta_t = \eta_t$, with $t$ taking discrete values $0, 1/M, 2/M, \ldots, 1$. For DRE-$\infty$, the coefficients are defined as $\alpha_t = \exp\{-0.25(\beta_{\max} - \beta_{\min})t^2 - 0.5\beta_{\min}t\}$ and $\beta_t = \sqrt{1 - \alpha_t^2}$ for the MNIST dataset and $\alpha_t = 1 - t$ and $\beta_t = t$ for other datasets. The corresponding stochastic process for the former one aligns with the solution to variance preserving (VP) SDEs (Song et al., 2021; Li et al., 2024b; Xin et al., 2024; Zhou et al., 2025b).

### B.2. From Optimal Transport to Entropic Regularization

The static OT problem seeks to find a coupling $\pi$ between two probability distributions $q_0$ and $q_1$ that minimizes a given cost function. For the 2-Wasserstein distance with a Euclidean ground cost $c(\mathbf{x}_0, \mathbf{x}_1) = \|\mathbf{x}_0 - \mathbf{x}_1\|^2$, the optimization problem is given by:

$$\mathcal{W}_2^2(q_0, q_1) = \inf_{\pi \in \Pi(q_0, q_1)} \int_{\mathbb{R}^d \times \mathbb{R}^d} \|\mathbf{x}_0 - \mathbf{x}_1\|^2 \mathrm{d}\pi(\mathbf{x}_0, \mathbf{x}_1), \tag{55}$$

where $\Pi(q_0, q_1)$ denotes the set of joint probability measures with marginals $q_0$ and $q_1$. The optimal solution for compactly supported distributions (Villani et al., 2009) is characterized by straight-line interpolations between samples:

$$\mathbf{X}_t = (1 - t)\mathbf{X}_0 + t\mathbf{X}_1, \quad t \in [0, 1], \tag{56}$$

where $\mathbf{X}_0 \sim q_0$ and $\mathbf{X}_1 \sim q_1$. This interpolation aligns with the Benamou-Brenier formulation (McCann, 1997; Zhou et al., 2024b), where the transport paths minimize the kinetic energy in the space of probability measures.

The natural connections between optimal transport theory and straight-line interpolations motivate the concept of Batch Optimal Transport (BatchOT) (Pooladian et al., 2023; Zhao et al., 2025). BatchOT provides a pseudo-deterministic coupling mechanism by extending the OT principles to minibatch sampling. This ensures practical scalability and aligns theoretical transport paths with computational requirements.

Despite its theoretical elegance, solving the OT problem at scale is computationally challenging due to its cubic complexity in the number of samples. Entropic regularization alleviates this issue by introducing an entropy penalty:

$$\mathcal{W}_{2,\xi}^2(q_0, q_1) = \inf_{\pi \in \Pi(q_0, q_1)} \int_{\mathbb{R}^d \times \mathbb{R}^d} \|\mathbf{x}_0 - \mathbf{x}_1\|^2 \mathrm{d}\pi(\mathbf{x}_0, \mathbf{x}_1) - \xi \mathcal{H}(\pi), \tag{57}$$

where $\xi > 0$ is the regularization parameter and $\mathcal{H}(\pi)$ denotes the entropy of $\pi$. This formulation ensures convexity and allows scalable computation via Sinkhorn's algorithm (Cuturi, 2013).

Entropic regularization connects OT with the Schrödinger bridge (SB) problem, which models stochastic interpolation between distributions. Given a reference Wiener process scaled by $\gamma$, the SB problem finds the most probable stochastic process $\pi$ that satisfies the marginal constraints $q_0$ and $q_1$:

$$\pi^\star = \underset{\pi \in \Pi(q_0, q_1)}{\arg\min} \ \mathrm{KL}(\pi \parallel \pi_{\mathrm{ref}}), \tag{58}$$

where $\pi_{\mathrm{ref}}$ is a reference process. The SB solution corresponds to an entropy-regularized OT plan with $\xi = 2\gamma^2$:

$$\mathbf{X}_t = (1 - t)\mathbf{X}_0 + t\mathbf{X}_1 + \sqrt{t(1 - t)\gamma^2}\mathbf{Z}_t, \tag{59}$$

where $\mathbf{Z}_t \sim \mathcal{N}(\mathbf{0}, \mathbf{E}_d)$. This formulation introduces stochasticity into the transport paths, effectively modeling uncertainty and noise.

## C. Experimental Details and More Results

### C.1. Comparison of the Trajectory Sets for Interpolation Strategies

In this section, we provide a detailed comparison of interpolation strategies, specifically deterministic interpolant (DI), diffusion bridge interpolant (DBI), dequantified diffusion bridge interpolant (DDBI), and dequantified Schrödinger bridge interpolant (DSBI). Their intermediate samples and corresponding distributions are visualized in Fig. 1.

**(a) DI:** DI constrains intermediate samples to fixed linear paths between $q_0(\mathbf{x})$ and $q_1(\mathbf{x})$, resulting in narrow bands across the trajectory space (Fig. 1(a)). While dense along the paths, DI severely limits support and fails to explore alternative trajectories, making it inflexible and unsuitable for diverse distributions.

**(b) DBI:** DBI introduces stochasticity through Brownian Bridge noise, expanding support and enabling broader trajectory exploration (Fig. 1(b)). Compared to DI, DBI provides greater coverage and variability while retaining tractability, reducing the rigidity of interpolation paths (Zhao et al., 2023).

**(c) DDBI:** Extending DBI, DDBI modulates the noise with deterministic interpolation weights and diffusion components. This results in more dispersed trajectories (Fig. 1(c)) and a larger coverage of the intermediate distributions, balancing controlled stochasticity with enhanced flexibility (Zhou et al., 2025a).

**(d) DSBI:** DSBI offers full stochastic control over noise and leverages entropy-regularized optimal transport, resulting in widely dispersed trajectories and efficient utilization of the trajectory space (Fig. 1(d)). By minimizing transition loss, DSBI achieves the largest support set and highest diversity among the methods, producing rich intermediate distributions.

Overall, our proposed methods (DBI, DDBI and DSBI) demonstrate clear advantages over deterministic baselines by achieving more comprehensive trajectory space exploration and flexible intermediate distribution generation. These results are consistent with our theoretical findings on support set and path set expansion, as formalized in Theorem 3.2 and Corollary 3.3.

## C.2. Joint Score Matching

In this section, we integrate the time score $s_{\boldsymbol{\theta}}^t \in \mathbb{R}$ and data score $\mathbf{s}_{\boldsymbol{\theta}}^{\mathbf{x}} \in \mathbb{R}^d$ to formulate the joint score $\mathbf{s}_{\boldsymbol{\theta}}^{t,\mathbf{x}} : [s_{\boldsymbol{\theta}}^t, \mathbf{s}_{\boldsymbol{\theta}}^{\mathbf{x}}] \in \mathbb{R}^{d+1}$. This joint score is incorporated into the training objective defined in Eq. (24), resulting in a joint score matching objective (Choi et al., 2022):

$$
\begin{aligned}
\mathcal{L}_4(\boldsymbol{\theta}) = {} & 2\mathbb{E}_{\mathbf{x}\sim q_0'(\mathbf{x})}[\lambda(0)\mathbf{s}_{\boldsymbol{\theta}}^{t,\mathbf{x}}(\mathbf{x},0)[t]] - 2\mathbb{E}_{\mathbf{x}\sim q_1'(\mathbf{x})}[\lambda(1)\mathbf{s}_{\boldsymbol{\theta}}^{t,\mathbf{x}}(\mathbf{x},1)[t]] \\
& + \mathbb{E}_{t\sim q(t)}\mathbb{E}_{\mathbf{x}\sim q_t'(\mathbf{x})}\mathbb{E}_{\mathbf{v}\sim q(\mathbf{v})}\left[2\lambda(t)\partial_t\mathbf{s}_{\boldsymbol{\theta}}^{t,\mathbf{x}}(\mathbf{x},t)[t] + 2\lambda'(t)\mathbf{s}_{\boldsymbol{\theta}}^{t,\mathbf{x}}(\mathbf{x},t)[t]\right. \\
& \left. + \lambda(t)\|\mathbf{s}_{\boldsymbol{\theta}}^{t,\mathbf{x}}(\mathbf{x},t)[\mathbf{x}]\|_2^2 + 2\lambda(t)\mathbf{v}^{\mathsf{T}}\nabla_{\mathbf{x}}\mathbf{s}_{\boldsymbol{\theta}}^{t,\mathbf{x}}(\mathbf{x},t)[\mathbf{x}]\mathbf{v}\right],
\end{aligned}
\tag{60}
$$

where $\mathbf{v} \sim q(\mathbf{v}) = \mathcal{N}(\mathbf{0}, \mathbf{E}_d)$ follows a standard Gaussian distribution, the terms $\mathbf{s}_{\boldsymbol{\theta}}^{t,\mathbf{x}}(\mathbf{x},t)[\mathbf{x}]$ and $\mathbf{s}_{\boldsymbol{\theta}}^{t,\mathbf{x}}(\mathbf{x},t)[t]$ represent the data and time score components of $\mathbf{s}_{\boldsymbol{\theta}}^{t,\mathbf{x}}(\mathbf{x},t)$, respectively.

## C.3. Mutual Information Estimation

Mutual information (MI) measures the dependency between two random variables $\mathbf{X} \sim p(\mathbf{x})$ and $\mathbf{Y} \sim q(\mathbf{y})$, quantifying how much information one variable contains about the other. In this experiment, we employ $D^3RE$ to estimate the MI between two $d$-dimensional correlated Gaussian distributions. Specifically, we consider $q(\mathbf{y}) = \mathcal{N}(\mathbf{0}, \sigma^2\mathbf{E}_d)$ and $p(\mathbf{x}) = \mathcal{N}(\mathbf{0}, \mathbf{E}_d)$, where $\sigma^2 = 1e-6$ and $d = \{40, 80, 120\}$. Let $p(\mathbf{x}, \mathbf{y})$ be the joint density of $\mathbf{X}$ and $\mathbf{Y}$. The MI between $\mathbf{X}$ and $\mathbf{Y}$ is defined as $\mathrm{MI}(\mathbf{X}, \mathbf{Y}) = \mathbb{E}_{\mathbf{x},\mathbf{y}\sim p(\mathbf{x},\mathbf{y})}\left[\log\frac{p(\mathbf{x},\mathbf{y})}{p(\mathbf{x})q(\mathbf{y})}\right]$, and can be approximated via DRE. We adapt the experimental setup of (Choi et al., 2022) to implement $D^3RE$.

To construct the joint distribution, we use $q_0(\mathbf{x}) = \mathcal{N}(\mathbf{0}, \mathbf{E}_d)$ and $q_1(\mathbf{x}) = \mathcal{N}(\mathbf{0}, \Sigma)$, where $\Sigma$ is block diagonal with $\Lambda = [[1, \rho], [\rho, 1]]$ as $2 \times 2$ sub-matrices. For $q_1$, it is designed as a multivariate normal distribution with a block diagonal covariance matrix along the block diagonal. Each $\Lambda$ represents the covariance between variable pairs, while off-diagonal blocks remain zero, ensuring no correlation across pairs. The DDBI and DSBI are implemented, given by $\mathbf{X}_t' = \alpha_t\mathbf{X}_0 + \beta_t\mathbf{X}_1 + \sqrt{t(1-t)\gamma^2 + (\alpha_t^2 + \beta_t^2)\varepsilon}\mathbf{Z}_t$, where $\mathbf{X}_0 \sim q_0(\mathbf{x})$, $\mathbf{X}_1 \sim q_1(\mathbf{x})$, and $\mathbf{Z}_t \sim \mathcal{N}(\mathbf{0}, \mathbf{E}_d)$. We estimate the density ratio $r(\mathbf{x}) = \frac{q_1(\mathbf{x})}{q_0(\mathbf{x})}$, yielding $\mathrm{MI}(\mathbf{X}, \mathbf{Y}) \approx \mathbb{E}_{\mathbf{x}\sim q_1(\mathbf{x})}[\log r(\mathbf{x})]$.

We train the score model using the joint score matching loss (details in Appendix C.2). The batch size is set to 512 for $d = \{40, 80, 160\}$ and 256 for $d = 320$, with iteration steps of $\{40k, 100k, 400k, 500k\}$, respectively. DRE-$\infty$ serves as the baseline method. Results, shown in Fig. 3, demonstrate that $D^3RE$, especially DSBI, produces MI estimates significantly closer and faster to the ground truth compared to the baseline, highlighting its superiority in accurately capturing mutual dependencies between variables.

## C.4. Density Estimation

**Energy-based Modeling on MNIST.** We applied the proposed $D^3RE$ framework to density estimation on the MNIST dataset, leveraging pre-trained energy-based models (EBMs) (Choi et al., 2022). Let $q_1(\mathbf{x})$ denote the MNIST data distribution and $q_0(\mathbf{x})$ a simple noise distribution with three different settings, as reported in (Rhodes et al., 2020): Gaussian noise, Gaussian copula, and Rational Quadratic Neural Spline Flow (RQ-NSF) (Durkan et al., 2019). We applied an modified version of DDBI of the form $\mathbf{X}'_t = \alpha_t \mathbf{X}_0 + \beta_t \text{EBM}(\mathbf{X}_1) + \sqrt{t(1-t)\gamma^2 + (\alpha_t^2 + \beta_t^2)\varepsilon}\mathbf{Z}_t$, where $\mathbf{X}_0 \sim q_0(\mathbf{x})$, $\mathbf{X}_1 \sim q_1(\mathbf{x})$, $\mathbf{Z}_t \sim \mathcal{N}(\mathbf{0}, \mathbf{E}_d)$, $\alpha_t = \exp\{-0.25(\beta_{\max} - \beta_{\min})t^2 - 0.5\beta_{\min}t\}$ and $\beta_t = \sqrt{1 - \alpha_t^2}$. $\beta_{\min}$ and $\beta_{\max}$ are set to 0.1 and 20, respectively. The results are reported in bits-per-dimension (BPD), evaluated as $\text{BPD} = -\frac{1}{d\ln 2}\mathbb{E}_{\mathbf{x}\sim q_1(\mathbf{x})}[\log q_1(\mathbf{x})]$, where the expectation reflects the log-density of the MNIST dataset. Exact BPD computation is infeasible for EBMs; therefore, we estimate it using two annealed MCMC methods: Annealed Importance Sampling (AIS) (Neal, 2001) and Reverse Annealed Importance Sampling Estimator (RAISE) (Burda et al., 2015).

*Table 3.* Comparison of the estimated log-density on MNIST dataset based on pre-trained energy-based models. The results are reported in BPD. Lower is better. The reported results for NCE and TRE are sourced from (Rhodes et al., 2020).

| Method | Noise type | Noise | Direct ($\downarrow$) | RAISE ($\downarrow$) | AIS ($\downarrow$) |
|---|---|---|---|---|---|
| NCE | Gaussian | 2.01 | 1.96 | 1.99 | 2.01 |
| TRE | Gaussian | 2.01 | 1.39 | 1.35 | 1.35 |
| DRE-$\infty$ | Gaussian | 2.01 | 1.33 | 1.33 | 1.33 |
| DRE-$\infty$+OTR, ours | Gaussian | 2.01 | 1.313 | 1.31 | 1.31 |
| $D^3RE$ (DDBI), ours | Gaussian | 2.01 | 1.297 | 1.30 | 1.29 |
| $D^3RE$ (DSBI), ours | Gaussian | 2.01 | **1.293** | 1.29 | 1.29 |
| NCE | Copula | 1.40 | 1.33 | 1.48 | 1.45 |
| TRE | Copula | 1.40 | 1.24 | 1.23 | 1.22 |
| DRE-$\infty$ | Copula | 1.40 | 1.21 | 1.21 | 1.21 |
| DRE-$\infty$+OTR, ours | Copula | 1.40 | 1.204 | 1.19 | 1.18 |
| $D^3RE$ (DDBI), ours | Copula | 1.40 | 1.193 | 1.19 | 1.19 |
| $D^3RE$ (DSBI), ours | Copula | 1.40 | **1.170** | 1.19 | 1.18 |
| NCE | RQ-NSF | 1.12 | 1.09 | 1.10 | 1.10 |
| TRE | RQ-NSF | 1.12 | 1.09 | 1.09 | 1.09 |
| DRE-$\infty$ | RQ-NSF | 1.12 | 1.09 | 1.08 | 1.08 |
| DRE-$\infty$+OTR, ours | RQ-NSF | 1.12 | 1.072 | 1.07 | 1.06 |
| $D^3RE$ (DDBI), ours | RQ-NSF | 1.12 | 1.072 | 1.06 | 1.06 |
| $D^3RE$ (DSBI), ours | RQ-NSF | 1.12 | **1.066** | 1.06 | 1.06 |

**2-D Synthetic Datasets.** In this section, we present density estimation results on eight synthetic datasets for different methods. From left to right, the epochs are 0, 2000, 4000, 6000, 8000, 10000, 12000, 14000, 16000, 18000 and 20000. Corresponding results are shown in Figs. 9 to 16. $D^3RE$ (including DDBI and DSBI) achieved the best performance on all datasets and was able to learn the best results with fewer epochs.

## C.5. Ablation Study on $\gamma^2$

**Mutual Information Estimation.** The ablation study on varying $\gamma^2$ values (Fig. 5) reveals distinct convergence behaviors in MI estimation across epochs. For all dimensions ($d = \{40, 80, 120\}$), smaller $\gamma^2$ values ($\leq 0.01$) lead to faster initial convergence toward the DRE-$\infty$ baseline, particularly in lower dimensions ($d = 40$). However, excessively small $\gamma^2 = 0.001$ introduces instability in later epochs, causing slight deviations from the baseline. In contrast, larger $\gamma^2$ values ($\geq 0.1$) show slower initial convergence but stabilize over longer training periods, especially in higher dimensions ($d = 120$). Notably, $\gamma^2 = 0.1$ strikes a balance between convergence speed and stability, consistently aligning with the baseline across all dimensions. These findings suggest that the optimal $\gamma^2$ selection is influenced by both the dimensionality and training duration, with moderate regularization ($\gamma^2 = 0.01$–$0.1$) providing robust MI estimation performance.

**Density Estimation.** The ablation study on $\gamma^2$ for density estimation (Fig. 4) reveals systematic trade-offs in performance across regularization strengths. For small $\gamma^2 = 0.001$, the model achieves rapid initial alignment with the ground truth distribution (first row) but exhibits overfitting artifacts in later epochs, manifesting as irregular density peaks and deviations from the smooth ground truth structure. Intermediate values ($\gamma^2 = 0.01$–$0.1$) demonstrate balanced behavior: $\gamma^2 = 0.01$ preserves finer details while maintaining stability, and $\gamma^2 = 0.1$ produces smoother approximations with minimal divergence from the true distribution. Larger $\gamma^2$ values ($\geq 0.5$) induce excessive regularization, leading to oversmoothed estimates that fail to capture critical modes of the 2-D data, particularly in high-density regions. Notably, $\gamma^2 = 0.1$ achieves the closest visual and structural resemblance to the ground truth, suggesting its suitability for low-dimensional tasks requiring both fidelity and robustness. These results underscore the necessity of tuning $\gamma^2$ to mitigate under-regularization artifacts while preserving distributional complexity.

We also present density estimation results on eight synthetic datasets for varing values of $\gamma^2$. From left to right, the epochs are 0, 2000, 4000, 6000, 8000, 10000, 12000, 14000, 16000, 18000 and 20000. Corresponding results are shown in Figs. 17 to 24.

### C.6. Ablation Study on OTR

Fig. 3 compares the MI estimation performance of DDBI and DSBI across different dimensions ($d = 80, 120$). DDBI uses diffusion bridges and Gaussian dequantization, while DSBI adds OTR to achieve better alignment. In panel (a), both DDBI and DSBI outperform the baseline methods. DDBI shows stable performance and converges close to the ground truth. However, DSBI, with OTR, achieves faster convergence and higher accuracy, staying closer to the ground truth throughout training. In panel (b), the impact of OTR becomes more evident. Although DDBI still outperforms the baseline methods, its convergence is slower, and its accuracy is lower compared to DSBI. By leveraging OTR, DSBI demonstrates superior MI estimation performance across all training epochs and remains closer to the ground truth in high-dimensional settings ($d = 120$). OTR significantly improves the alignment of intermediate distributions and enhances model performance. When combined with diffusion bridges and Gaussian dequantization, as in DSBI, OTR achieves its full potential. It allows the model to estimate complex distributions more accurately.

**Number of Function Evaluations.** We analyze the impact of OTR on NFE, noting that DI and DDBI do not utilize OTR. Our observations show that applying OTR significantly reduces NFE. Fig. 8 compares NFE across four methods in DRE, highlighting substantial variations in computational efficiency. The first approach exhibits the highest NFE, indicating reliance on iterative procedures requiring repeated function evaluations. The second approach achieves a moderate reduction in NFE, likely by minimizing redundant evaluations through minimized transport costs.

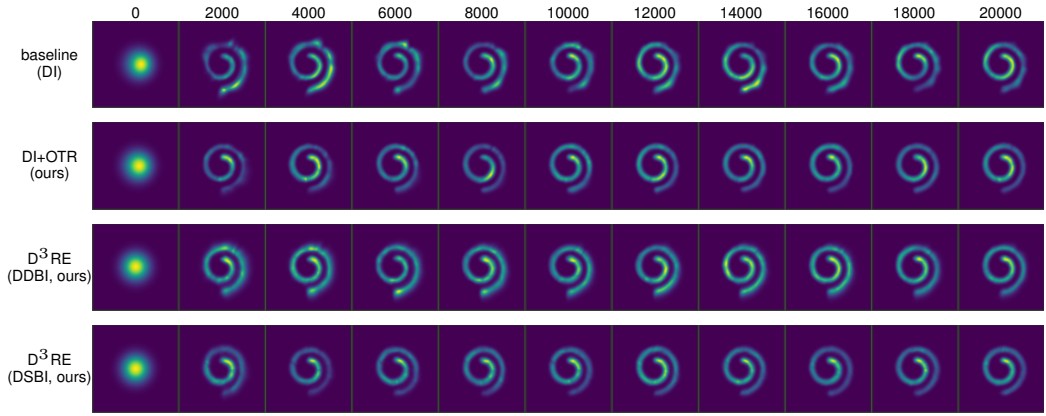

*Figure 9.* Density estimation results on swissroll for different methods during training.

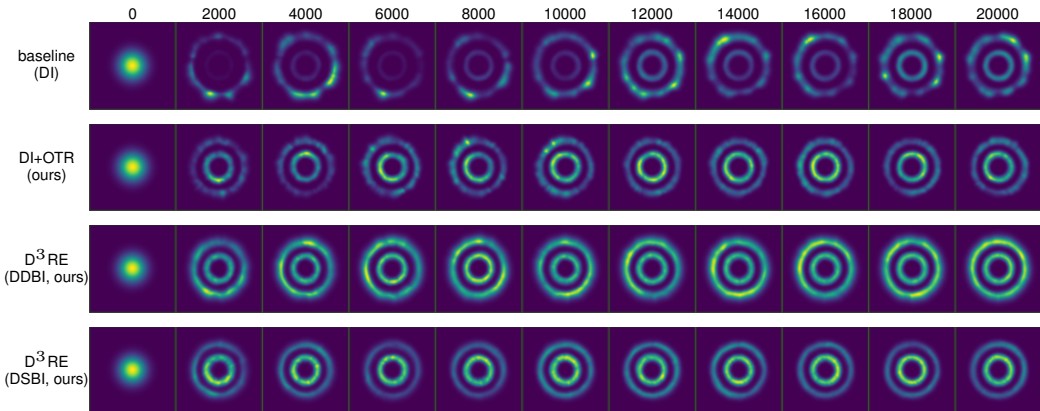

*Figure 10.* Density estimation results on circles for different methods during training.

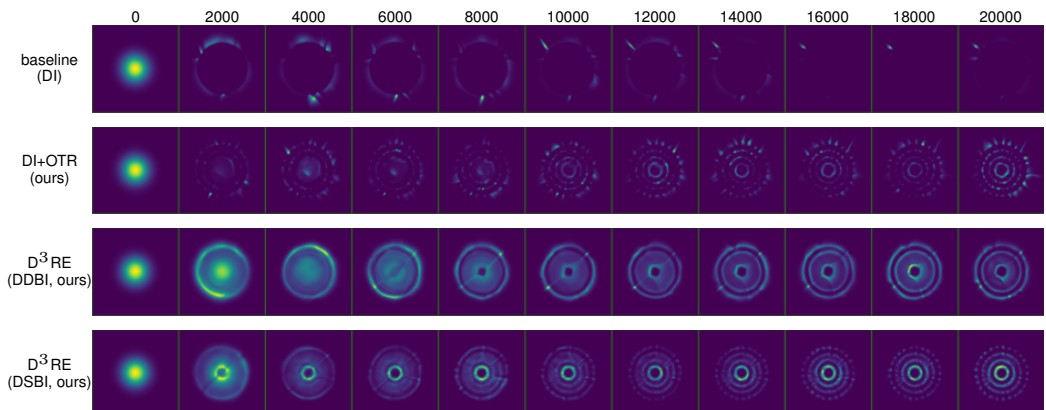

*Figure 11.* Density estimation results on rings for different methods during training.

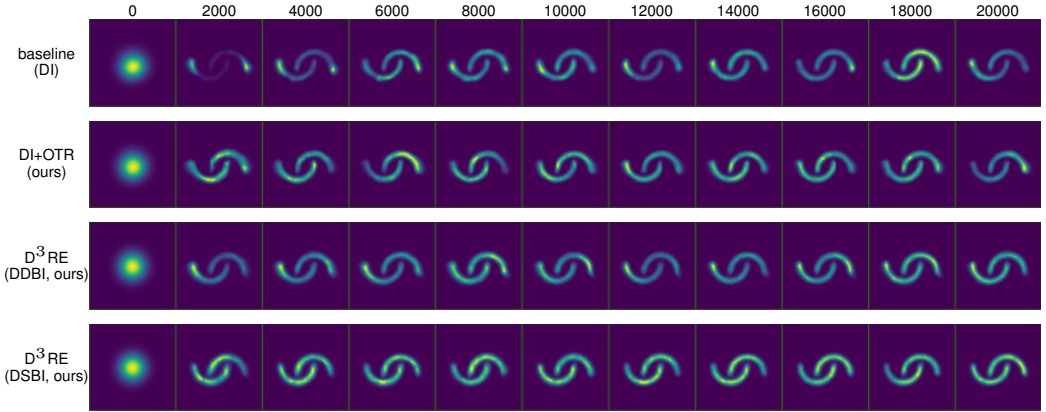

*Figure 12.* Density estimation results on moons for different methods during training.

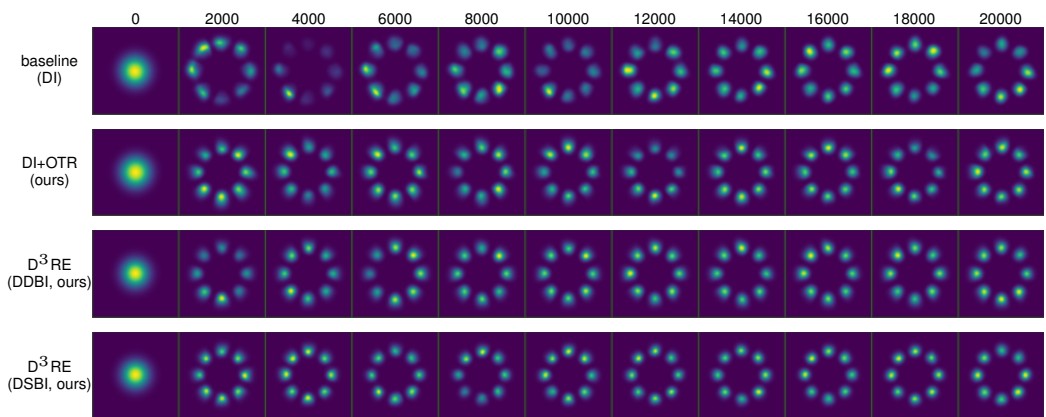

*Figure 13.* Density estimation results on 8gaussians for different methods during training.

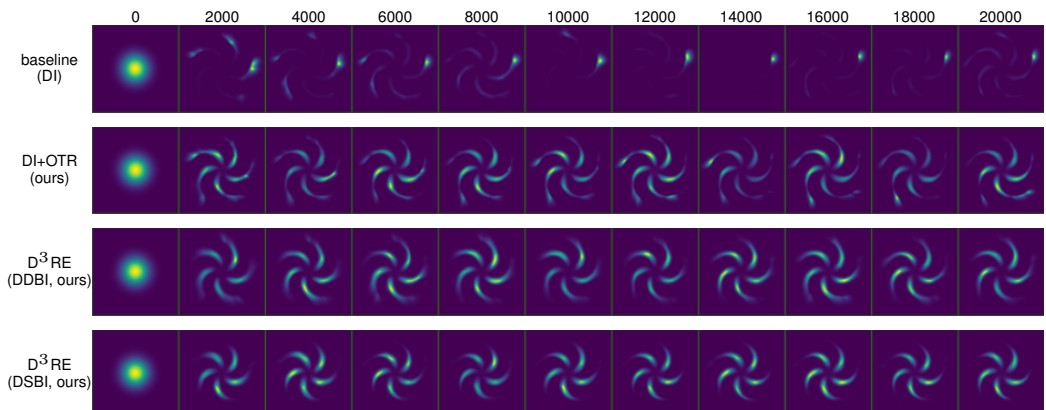

*Figure 14.* Density estimation results on pinwheel for different methods during training.

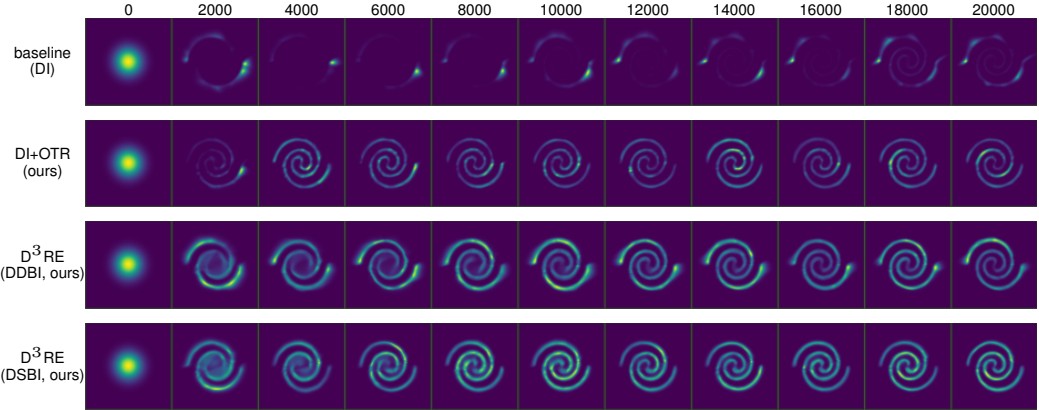

*Figure 15.* Density estimation results on 2spirals for different methods during training.

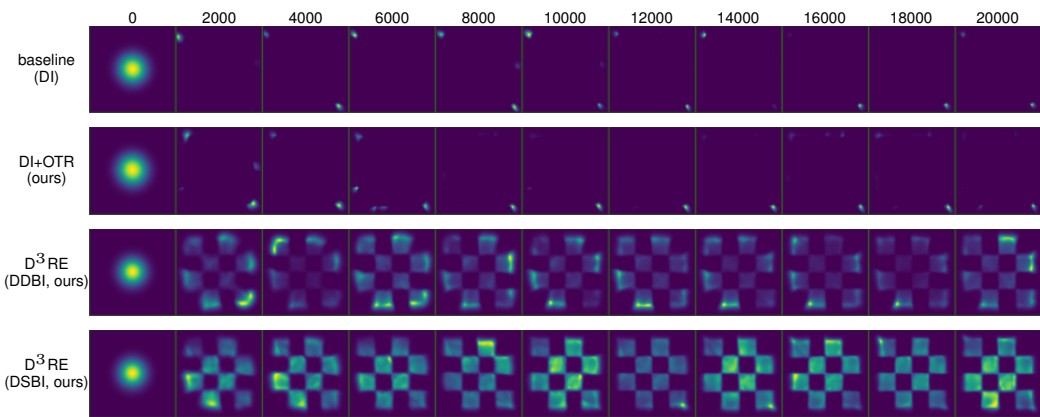

*Figure 16.* Density estimation results on checkerboard for different methods during training.

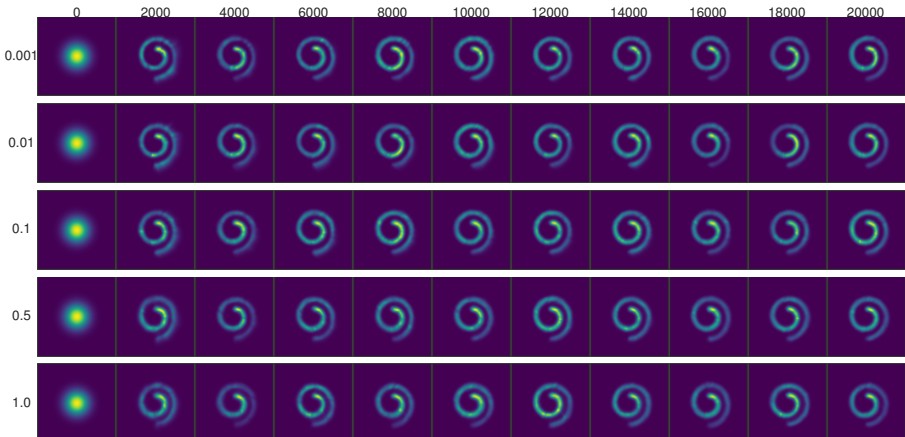

*Figure 17.* Density estimation results on swissroll for varing values of $\gamma^2$ during training.

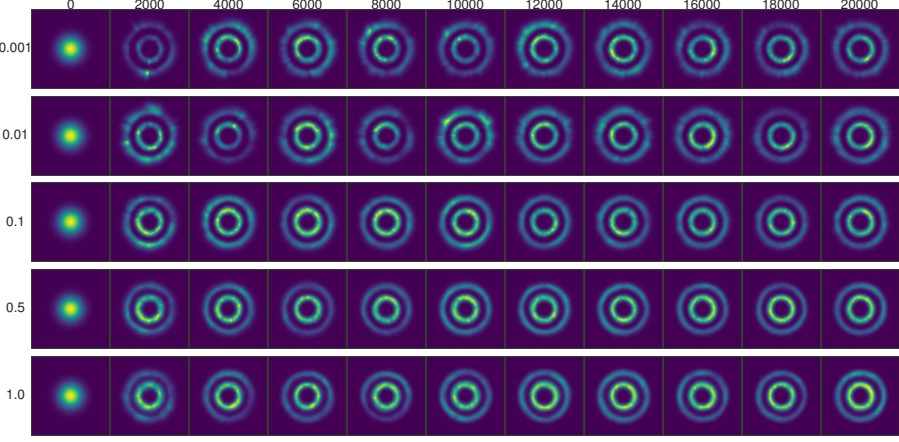

*Figure 18.* Density estimation results on circles for varing values of $\gamma^2$ during training.

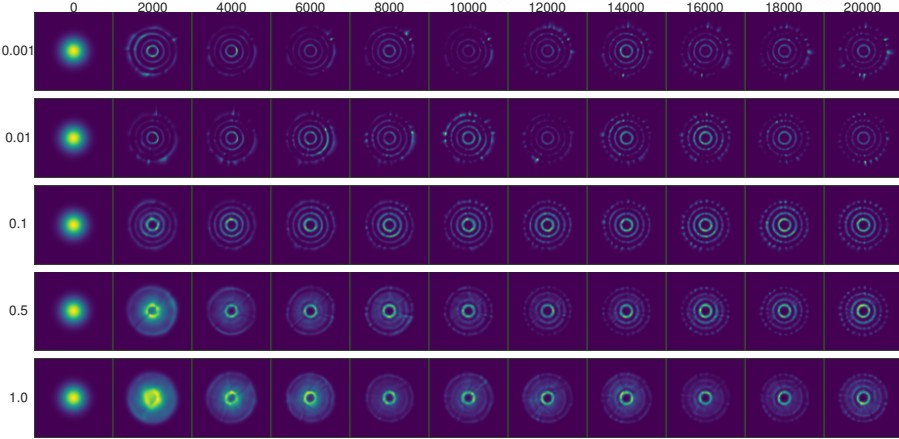

*Figure 19.* Density estimation results on rings for varing values of $\gamma^2$ during training.

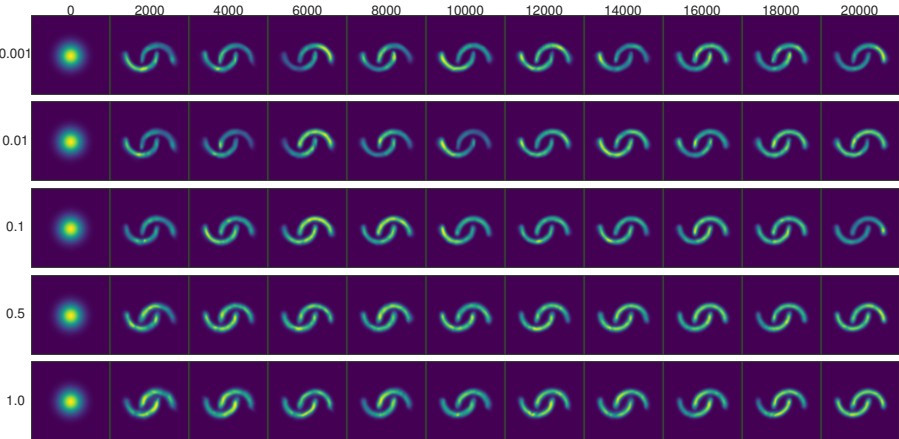

*Figure 20.* Density estimation results on moons for varing values of $\gamma^2$ during training.

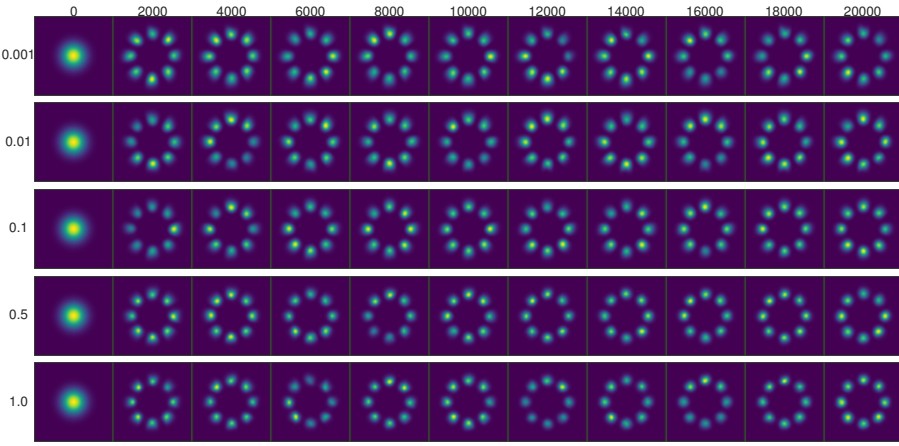

*Figure 21.* Density estimation results on 8gaussians for varing values of $\gamma^2$ during training.

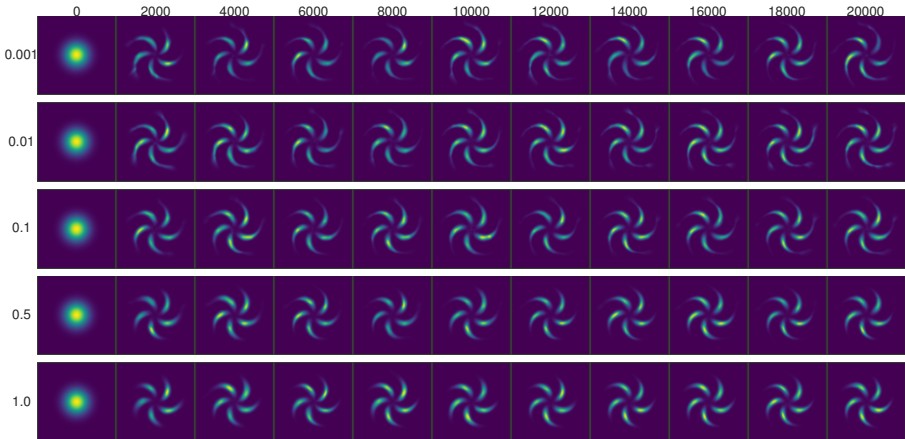

*Figure 22.* Density estimation results on pinwheel for varing values of $\gamma^2$ during training.

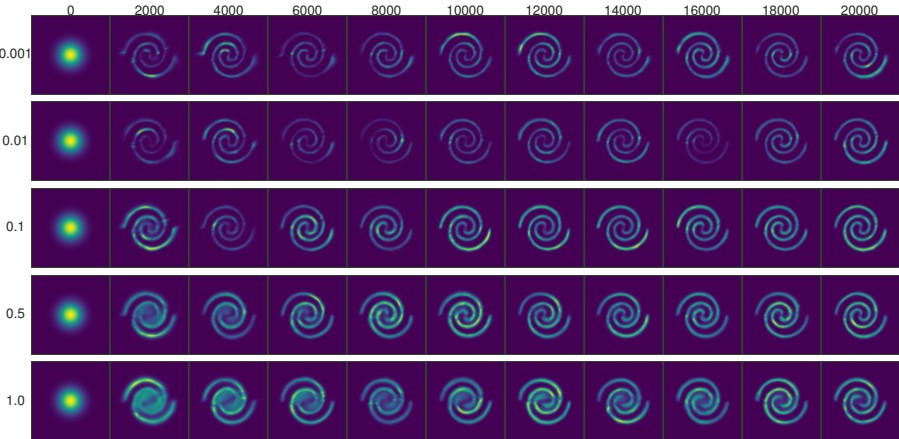

*Figure 23.* Density estimation results on 2spirals for varing values of $\gamma^2$ during training.

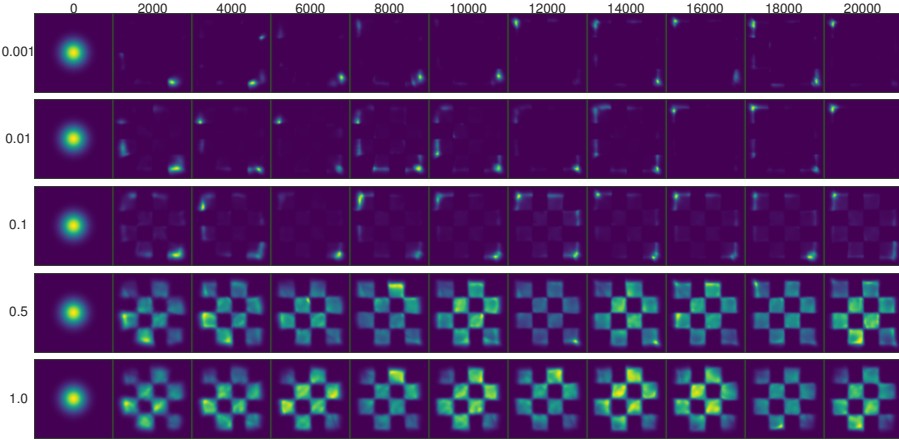

*Figure 24.* Density estimation results on checkerboard for varing values of $\gamma^2$ during training.

