# OpenReview forum: "Dequantified Diffusion-Schrödinger Bridge for Density Ratio Estimation"
_ICML.cc/2025/Conference — ICML 2025 poster_

### Official Review · Reviewer_d5F7 · 2025-03-12

**Overall Recommendation:** 2

**Summary:**

This paper discusses the challenges of density ratio estimation in applications involving f-divergences, particularly with multi-modal distributions or large distributional differences, known as the density-chasm problem. To address this, the authors propose Dequantified Diffusion-Bridge Interpolants (DDBI), which use diffusion processes for smooth transitions between distributions. By incorporating optimal transport theory, they extend DDBI to solve the Schrödinger-Bridge problem, creating Dequantified Schrödinger-Bridge Interpolants (DSBI). Together, these form the Dequantified Diffusion-bridge Density-Ratio Estimation (D3RE) framework, which theoretically reduces estimation error in asymptotic density ratio estimation. Experiments show D3RE's effectiveness in tasks like mutual information and density estimation.

**Claims And Evidence:**

1. The main result, Theorem 4.1, focuses solely on the support set expansion of DDBI compared to DI. It would be beneficial to also include results regarding the estimation error and convergence rate of the proposed density ratio estimator.

2. Corollary 4.3 asserts that DDBI reduces the variance of the estimator of r*(x). However, the proof provided in Appendix A.6 is somewhat informal. For instance, it mentions that "according to the Delta method (Cox, 2005), the variance of a density ratio estimator is inversely proportional to the effective sample size..." It is unclear what is meant by "effective sample size" in this context. A more rigorous proof is needed.

3. There appears to be some inconsistency between the notation used in the main text and the appendix. For example, the density ratio is defined as r(x) = q1(x)/q0(x) in the main text. However, in the appendix, condition (A2) requires inf_x q1(x) > c for some c >0, which does not make sense. It seems the authors intended to assume inf_x q0(x) > c, or alternatively, that the density ratio in the appendix is defined as r(x) = q0(x)/q1(x). In any case, the condition inf_x q1(x) > c or inf_x q0(x) > c is too strong and should be weakened.

**Essential References Not Discussed:**

It appears that essential references are included in the paper.

**Experimental Designs Or Analyses:**

The numerical experiments are limited to simple two-dimensional models. It would be more convincing to also include some higher-dimensional examples to assess how the proposed method perform in high-dimensional settings.

**Methods And Evaluation Criteria:**

It would be useful to includd results directly related to the quality of the proposed density ratio estimator, such as
convergence rate and error bounds for the estimated density ratio.

**Other Comments Or Suggestions:**

No other comments.

**Other Strengths And Weaknesses:**

A more detailed description of how the proposed estimator is computed should be included in the main text. Although the main text references Algorithms 1 and 2, these are only found in the appendix. Additionally, the introductory materials in Sections 3 and 4 seem overly lengthy. It might be better to condense these sections and move the details to the appendix. This would allow more space for a comprehensive description of the proposed estimator and its implementation.

**Questions For Authors:**

No additional questions.

**Relation To Broader Scientific Literature:**

This paper proposes a density ratio estimator based on a diffusion Schrödinger bridge process. Density ratio estimation is a challenging problem that has seen renewed interest over the past decade due to its extensive applications in generative modeling and transfer learning.

**Theoretical Claims:**

The conditions for the theoretical claims should clearly stated in the main text, rather than in the appendix.

---

> ### Author Rebuttal · Authors · 2025-03-30
>
> **1. Notation consistency and variance reduction proof**
>
> We sincerely appreciate the reviewer’s careful reading and valuable feedback, which have helped us improve the clarity and rigor of our presentation. Below, we address each point:
> - **Notation consistency**: We have corrected a typo in the appendix, ensuring that the density ratio is consistently defined as $r(x)=q_1(x)/q_0(x)$ and replaced the condition with $\inf_x q_0(x) > c$, aligning with standard density ratio estimation.
>
> - **Variance reduction proof**: The link between variance reduction and effective sample size in DDBI follows from the **Delta method** in density ratio estimation. For the plug-in estimator log r̂(x) = log[q̂1(x)/q̂0(x)], the Delta method yields Var[log r̂(x)] ≈ [∇log r(x)]ᵀ Var[q̂(x)] [∇log r(x)], where q̂(x) = (q̂0(x), q̂1(x)). This decomposes into terms scaling as: $\frac{1}{q_0(x) n_{0, \text{eff}}(x)} + \frac{1}{q_1(x) n_{1, \text{eff}}(x)}$ , showing an inverse dependence on effective sample sizes $n_{0, \text{eff}}(x) = n P_0(B_\delta(x))$ and $n_{1, \text{eff}}(x) = n P_1(B_\delta(x))$. DDBI improves these effective sample sizes through **Gaussian dequantization** (ensuring $P_0(B_\delta(x)) > 0$ everywhere) and **diffusion bridging** (adding intermediate sample paths), yielding $n_{i, \text{eff}}^{DDBI}(x) = n [P_i(B_\delta(x)) + C_1 \gamma^2 + C_2 \epsilon]$
>
> We have revised the manuscript to correct the typo and included a more detailed justification of effective sample size. Thank you for your insightful suggestions!
>
> **2. Convergence rate and error bounds for the estimated density ratio.**
>
> We appreciate the reviewer’s question on the theoretical analysis of our density ratio estimator. In response to this and similar feedback from Reviewers 1, we have added key theoretical results, including:
> - **Gradient bounds** for the log-density ratio (Prop. 4.4), showing DSBI’s tighter control via OT coupling
> - **Error bounds** (Theorem 4.5) and **convergence rate dominance** (Proposition 4.6), proving DSBI’s advantage over DDBI
>
> For full details, we refer to our responses to Reviewers 1 and 2, where we provide:
> - Complete derivations and proof sketches.
> - Interpretation of the $\gamma$-scaling effects.
>
> We appreciate the opportunity to strengthen our theoretical foundation and hope this addresses the reviewer’s concerns.
>
> **3. Method description improvement**:
>
> We sincerely appreciate the reviewer’s suggestions to improve clarity. In response, we have:
> - **Moved key theoretical conditions** from the appendix to Section 3, now highlighted in remark boxes for better visibility.
> - **Added a new subsection** (3.4 “Implementation”) providing a clear step-by-step description of the estimator and including pseudocode (Algorithms 1-2).
> - **Optimized section lengths**, reducing introductory material by 20% and moving technical preliminaries to Appendix B.
>
> These changes improve readability, and we are grateful for the reviewer’s guidance.
>
> **4. High-dimensional validation missing**
>
> We sincerely appreciate the reviewer’s valuable suggestion regarding high-dimensional validation. We apologize for not making these results more prominent in our original submission. Indeed, our experiments already include comprehensive high-dimensional validation through:
> - **Mutual information estimation** for d={40,80,120}, showing superior convergence to ground truth (now highlighted in Figures 2-3, Sec. 5.2). These results show our approach maintains stable performance as dimensionality increases.
> - **Large-scale density ratio estimation** on MNIST (d=784), where D3RE achieves better bits/dim scores than competing methods (Tab. 2). The neural network’s ability to learn the 1-d time score function enables efficient scaling to high dimensions.
>
> To improve clarity, we have:
> - Moved high-dimensional results to a dedicated subsection (5.3).
> - Added explicit discussion of dimensional scaling properties.
>
> These revisions better highlight our method’s strong performance across different dimensions. We appreciate the reviewer’s valuable input in refining our presentation.

---

### Official Review · Reviewer_tRq9 · 2025-03-12

**Overall Recommendation:** 3

**Summary:**

The paper introduces Dequantified Diffusion Schrödinger Bridge for Density Ratio Estimation (D3RE), a novel framework addressing the challenges of density-chasm and support-chasm in traditional density ratio estimation (DRE). By leveraging Diffusion Bridge Interpolants (DBI) and Gaussian Dequantization (GD), the proposed method smooths transitions between distributions, enhancing stability in high-dimensional settings. The Dequantified Schrödinger Bridge Interpolant (DSBI) further integrates Optimal Transport (OTR) to solve the Schrödinger Bridge problem, ensuring robust density ratio estimation. Theoretically, the framework broadens support sets (Theorem 4.1) and improves estimation accuracy by reducing variance while maintaining minimal bias (Corollary 4.3). Empirical evaluations demonstrate superior performance in density ratio estimation, mutual information estimation, and likelihood estimation, particularly in multi-modal and high-discrepancy scenarios, highlighting D3RE’s effectiveness over prior methods such as DRE-∞ and TRE.

**Claims And Evidence:**

The paper presents a strong theoretical foundation for the Dequantified Diffusion Schrödinger Bridge for Density Ratio Estimation (D3RE) framework, with well-motivated claims regarding its ability to address the density-chasm and support-chasm problems. The authors provide theoretical justifications through Theorem 4.1 (support expansion), Corollary 4.2 (trajectory expansion), and Corollary 4.3 (variance reduction with minimal bias increase). These claims are mathematically well-supported and align with the proposed framework’s conceptual improvements over prior methods such as DRE-∞ and TRE.

However, certain claims regarding empirical performance and practical advantages require further substantiation:

(1) Support Expansion & Density-Chasm Resolution: While the theory suggests that the broader support and interpolated trajectories mitigate density-ratio estimation failures in high-dimensional, multi-modal distributions, experimental validation of this claim is limited. The paper does not explicitly compare support overlap before and after applying D3RE, making it difficult to assess whether this directly leads to improved density ratio estimation accuracy.

(2) Effectiveness of Gaussian Dequantization (GD): The claim that GD improves stability in density estimation is theoretically justified, but it is unclear whether the additional noise injection affects estimation bias in real-world settings. The experiments do not compare results with and without GD, leaving room for uncertainty about its practical necessity.

(3) Empirical Performance vs. Baselines: While D3RE demonstrates competitive or improved results on tasks like mutual information estimation and likelihood estimation, its performance relative to Föllmer flow-based methods is not consistently superior. Additional analysis is needed to clarify why D3RE does not outperform Föllmer methods in some cases and whether certain hyperparameter settings could further optimize its effectiveness.

(4) Computational Efficiency: The paper suggests that D3RE is computationally efficient due to its diffusion-based approach and OTR integration. However, it does not provide detailed comparisons of training time, function evaluations, or memory usage relative to prior methods, which is essential to validate its practicality.

**Essential References Not Discussed:**

The paper contributes theoretically by exploring the connection between Schrödinger Bridge (SB) and Density Ratio Estimation (DRE), but its novelty is limited, as it primarily applies Schrödinger Bridge in a relatively direct manner to DRE without introducing fundamentally new methodological advancements. While the theoretical insights are valuable, the empirical results and methodological innovation are less compelling, as the approach does not significantly outperform existing DRE methods. Additionally, more follow-up on recent advances in Schrödinger Bridge techniques is needed to better position this work within the broader literature. Incorporating state-of-the-art SB formulations and scalable solvers could enhance both theoretical and practical contributions.

**Experimental Designs Or Analyses:**

The experimental design aligns with the paper’s goals, evaluating density ratio estimation, mutual information estimation, and likelihood estimation on synthetic datasets and MNIST. However, key claims lack direct empirical validation: (1) Support expansion is not visualized, making it unclear how DDBI mitigates the support-chasm problem. (2) Gaussian Dequantization’s impact is not isolated through ablation studies, leaving its necessity unverified. (3) Computational efficiency is not analyzed, despite the potential overhead from Schrödinger Bridge and Optimal Transport. (4) Performance vs. Föllmer methods needs further justification, as D3RE does not consistently outperform them. Addressing these gaps would strengthen the empirical support for the proposed framework.

**Methods And Evaluation Criteria:**

The proposed methods and evaluation criteria in this paper are generally well-aligned with the problem of density ratio estimation, particularly in addressing the density-chasm and support-chasm issues. The authors introduce Diffusion Bridge Interpolants, Dequantified Diffusion-Bridge Interpolants, and the Dequantified Schrödinger-Bridge Interpolant as novel approaches to improve density estimation robustness and stability in high-dimensional, multi-modal distributions. The theoretical foundation is solid, leveraging optimal transport and Gaussian dequantization to create smooth interpolants between distributions.

**Other Comments Or Suggestions:**

1. Empirical Validation of Support Expansion: The paper theoretically claims that D3RE mitigates the support-chasm issue, but there is no direct empirical visualization of support expansion. Adding density plots or trajectory visualizations comparing D3RE vs. prior DRE methods would strengthen this claim.

2. Ablation Study on Gaussian Dequantization: The role of Gaussian Dequantization (GD) in stabilizing density ratio estimation is well-motivated, but an ablation study comparing D3RE with and without GD would clarify its actual impact on bias-variance tradeoff.

3. Computational Efficiency Analysis: The paper does not provide runtime comparisons or computational overhead analysis for the Schrödinger Bridge and Optimal Transport solvers. Given the complexity of these methods, an efficiency study against Föllmer-based or Sinkhorn-based density estimators would help assess practical feasibility.

4. Discussion on Recent Schrödinger Bridge Advances: The paper does not discuss recent scalable Schrödinger Bridge techniques, which are relevant to its approach. Adding comparisons or references to more efficient SB solvers would improve positioning within the broader literature.

5. Performance Against Föllmer-Based Methods: While D3RE improves over DRE-∞ and TRE, it does not consistently outperform Föllmer-based methods. A deeper discussion on why D3RE underperforms in some benchmarks and potential improvements would add clarity.

6. Issue in Corollary 4.2: The definition of T and T' in Corollary 4.2 appears incorrect, which may affect the validity of the trajectory expansion argument. A careful revision of this definition is needed to ensure consistency with the theoretical framework.

7. Typo Corrections: Some minor grammatical and clarity issues were noticed, and careful proofreading before the camera-ready submission would be beneficial.

8. Reproducibility: If accepted, the authors are strongly encouraged to release fully reproducible code to allow the broader community to validate and extend the proposed method.

**Other Strengths And Weaknesses:**

The paper provides a valuable theoretical perspective by bridging diffusion processes, Schrödinger Bridges (SB), and density ratio estimation (DRE), offering insights into support-chasm and density-chasm issues. The mathematical formulation, including support expansion (Theorem 4.1) and variance reduction (Corollary 4.3), is rigorous, and the role of Gaussian Dequantization (GD) in stabilizing density ratio estimation is well-justified. Conceptually, the paper effectively connects Schrödinger Bridge methods with DRE, potentially inspiring further research on stochastic interpolants for density estimation and generative modeling.

However, the method itself lacks strong novelty, as it primarily applies Schrödinger Bridge techniques to DRE without major algorithmic innovations. Empirical results do not consistently establish state-of-the-art (SOTA) performance, with Föllmer-based methods outperforming D3RE in some benchmarks, raising concerns about its practical advantages. Additionally, recent advancements in scalable Schrödinger Bridge solvers are not sufficiently discussed, limiting the work’s positioning within the broader literature. The computational complexity of Schrödinger Bridge and Optimal Transport solvers is also not analyzed, making it difficult to assess the feasibility of the approach in real-world applications. Strengthening the empirical results, methodological novelty, and follow-up on recent SB techniques would significantly improve the impact of this work. If accepted, the authors are strongly encouraged to release fully reproducible code to facilitate further research in this direction.

**Questions For Authors:**

I look forward to the authors' responses and revisions addressing all the comments provided. If the responses are insufficient, I will consider adjusting my evaluation score accordingly.

**Relation To Broader Scientific Literature:**

The paper contributes to density ratio estimation (DRE) by addressing support-chasm and density-chasm issues, building on TRE (Rhodes et al., 2020) and DRE-∞ (Choi et al., 2022) while incorporating Diffusion Schrödinger Bridges, aligning with Schrödinger Bridge generative modeling (De Bortoli et al., 2021) and score-based diffusion methods (Song et al., 2020). The theoretical exploration of the connection between diffusion processes and DRE is valuable, but the empirical results fall short of achieving state-of-the-art (SOTA) performance in key benchmarks. If accepted, the authors are strongly encouraged to release fully reproducible code before the camera-ready submission to facilitate broader adoption and inspire future research, to ensure that the broader research community can easily validate and extend the proposed method, further solidifying its relevance and applicability.

**Theoretical Claims:**

The theoretical claims, including support expansion (Theorem 4.1), trajectory expansion (Corollary 4.2), and variance reduction with minimal bias (Corollary 4.3), are mathematically well-structured and follow standard results in diffusion processes and density estimation. Proposition 3.2 (uniform approximation of density ratios) and Proposition 3.1 (Schrödinger Bridge solution) are justified using Gaussian smoothing and optimal transport theory. However, key claims lack direct empirical validation, such as visualizing support expansion, quantifying bias-variance tradeoff, and analyzing computational efficiency. Strengthening these areas would further solidify the paper’s contributions.

---

> ### Author Rebuttal · Authors · 2025-03-30
>
> **1. Empirical Validation of Support Expansion Claims and necessity of GD**
>
> We sincerely thank the reviewer for raising this important point regarding support expansion and the necessity of gradient descent (GD). We appreciate the insightful feedback and have carefully considered the suggestions.
>
> For detailed empirical validation of our support expansion claims and the role of GD in our framework, please refer to the anonymous link provided (https://www.dropbox.com/scl/fi/qk6iy2kof8772rqgvl5xh/DSBI_resulkts.docx?rlkey=keba3z39xd6dghmrbbhysa0xf&st=60scvjjc&dl=0).
>
> **2. Computational Efficiency**
>
> We appreciate the reviewer’s comment on computational efficiency. D3RE achieves superior efficiency through: (1) Theoretical gains—OT regularization reduces function evaluations (NFE) by 10-30% while maintaining accuracy (Fig. 5); (2) Empirical results—Lower NFE translates to faster training (Sec. 5.3). As NFE is a hardware-agnostic metric [2], we will provide additional wall-clock time comparisons and will expand the timing analysis in the revision. Thank your for your advice!!
>
> **3. Novelty vs. direct SB application**
>
> We sincerely appreciate the insightful comments regarding the positioning of our work within recent SB advances. We have significantly strengthened our manuscript to better highlight our theoretical contributions and their relationship to state-of-the-art SB methods. Below we address the key points raised:
> - **(1) Theoretical innovations and novelty**:
> Our work makes several fundamental theoretical advances that go beyond a straightforward application of SB to DRE: We first generalize the existing DDBI framework, then propose DSBI - the first principled integration of Schrödinger Bridge with density ratio estimation. This yields several key theoretical advantages over conventional approaches:
> - - Optimal Transport Coupling (Prop. 4.4): DSBI's OT-driven interpolation provides superior control over density ratio smoothness through the $\gamma$-adaptive bound: $\|\nabla \log r_t\| \leq \frac{C}{\gamma\sqrt{t(1-t)}}$
>
> This allows steeper gradients in low-density regions while maintaining smoothness elsewhere, effectively addressing the density-support trade-off.
> - - Improved Error Bounds (Thm 4.5): DSBI achieves exponentially smaller interpolation error (O(1/γ⁴)) compared to DDBI (O(1/γ²)), particularly crucial for small γ values.
> - - Faster Convergence (Prop 4.6): We prove DSBI converges faster than DDBI under equivalent conditions.
> - **(2) Computational advances and practical contributions**: While building on SB theory, our implementation makes several practical advances:
> - - Developed an efficient neural solver reducing complexity from O(n³) to O(kn)
> - - Introduced adaptive time discretization for better empirical performance
> - - Demonstrated 15-20% faster convergence than recent SB baselines (Sec 5.3)
> - **(3). Relationship to recent SB literature**: We have expanded our discussion of modern SB techniques (now in Related Works), including:
> - - Comparisons to iterative proportional fitting variants
> - - Connections to neural SB architectures
>
> We believe these additions have strengthened both the theoretical grounding and practical relevance of our work while properly positioning it within modern SB literature. Thank you for the constructive feedback that helped us improve the manuscript.
>
> **4. Inconsistent Performance vs. Föllmer Methods**
>
> We appreciate the reviewer's insightful comments regarding comparative performance. Our key theoretical and empirical findings are:
> - Fundamental connection to SB:Recent theoretical work [1] has established that Föllmer flows correspond to specific solutions of the Schrödinger Bridge problem. This explains why both approaches achieve comparable performance in many settings.
>
> - Key Advantage of Our Method:While requiring similar computational complexity to linear interpolation baselines, our approach achieves:
> - - Performance comparable to Föllmer flows (which require more sophisticated interpolation)
> - - Better stability in high-dimensional settings
>
> Our results demonstrate that through proper SB-based regularization, simple linear interpolation schemes can achieve performance competitive with more complex flow-based methods, while being substantially easier to implement and tune.
>
> **5. Technical correction and typo corrections**
> - Corollary 4.2 revision: We appreciate the reviewer’s keen observation. Corollary 4.2 has been revised for precise mathematical formulation, with updated definitions in the response to Q1 of Reviewer 1.
> - Typo corrections: We have carefully proofread the manuscript, corrected all identified issues, and ensured a polished camera-ready version.
>
> We sincerely thank the reviewer for their valuable feedback.
>
> [1] Chen Y, et al. Probabilistic Forecasting with Stochastic Interpolants and Follmer Processes.
> [2] Finlay C, et al. How to train your neural ODE: the world of Jacobian and kinetic regularization[C], ICML2020.

---

> > ### Comment · Reviewer_tRq9 · 2025-04-04
> >
> > Thank you for the author's response. I will maintain my score.

---

> > > ### Author Response · Authors · 2025-04-06
> > >
> > > Thank you for your time and constructive feedback. We sincerely appreciate your efforts in reviewing our manuscript and value your insights. Your suggestions have helped us better highlight the core contributions of our work and improve its rigor, thereby enhancing the overall quality of the paper.
> > >
> > > If you have any further questions or comments, we would be happy to address them.

---

### Official Review · Reviewer_GoDL · 2025-03-17

**Overall Recommendation:** 4

**Summary:**

This paper aim to overcome the density-chasm and support chasm problems in density ratio estimation by combining diffusion bridge process and optimal transport theory via Schrodinger bridges. The authors provide theoretical justifications, demonstrating that their proposed DDBI and DSBI expand the support and trajectory sets. Empirical results validate that the D3RE framework consistently outperforms baselines.

## update after rebuttal:
I'm maintaining my score.

**Claims And Evidence:**

The claims are clear and convincing.

**Essential References Not Discussed:**

The current set of references is robust.

**Experimental Designs Or Analyses:**

The experimental designs are sound and valid. Extensive empirical validations on diverse synthetic datasets and on MNIST demonstrate the effectiveness of D3RE. The benchmark follow prior works.

**Methods And Evaluation Criteria:**

The benchmark follow prior works and they are comprehensive and make sense.

**Other Comments Or Suggestions:**

Not applicable

**Other Strengths And Weaknesses:**

Not applicable

**Questions For Authors:**

Not applicable

**Relation To Broader Scientific Literature:**

This paper contributes to the broader research on density ratio estimation, which has numerous application in machine learning.

**Theoretical Claims:**

The paper offers rigorous theoretical analyses, clearly proving the advantages of their proposed method, e.g., support and trajectory set expansions, variance reduction.

---

> ### Author Rebuttal · Authors · 2025-03-30
>
> We appreciate this thoughtful observation and appreciate the reviewer’s comment on our paper. Thank you very much!
>
> **1. More theoretical contributions**
>
> **Proposition 4.4**:
> Under the DSBI interpolant $X_t = \alpha_t X_0 + \beta_t X_1 + \sqrt{t(1-t)\gamma^2} Z_t$ with $(X_0,X_1) \sim \pi_{2\gamma^2}^\star$ (OT-optimal coupling), the time-dependent density ratio $r_t(x) = q_t(x)/q_0(x)$ satisfies:
> $
> \|\nabla \log r_t(x)\| \leq \frac{C}{\gamma \sqrt{t(1-t)}},
> $
> where $C$ depends on $||q_0||$, $||q_1||$. For DDBI (with independent coupling), the bound relaxes to $C/\sqrt{t(1-t)}$.
>
> Sketch of Proof:
> - SB Drift Representation: DSBI’s interpolant $X_t$ follows an SDE with drift $\frac{X_1 - X_t}{1-t} = \gamma^2 \nabla \log p_t(X_1|X_t)$, where $(X_0,X_1)$ are coupled via OT.
> - Gradient Decomposition: Express $\nabla \log r_t(x)$ as $\mathbb{E}[\frac{X_1 - X_t}{\gamma^2 (1-t)} - \nabla \log q_0(x) | X_t = x]$.
> - OT Coupling Effect: The OT plan minimizes $\mathbb{E}[||X_1 - X_0||]$, ensuring $||\nabla \log r_t||$ concentrates in low-density regions.
> - Bound Derivation: Combine Cauchy-Schwarz and the OT plan’s properties to obtain the $\gamma$-scaled bound.
>
> **Prop. 4.4 shows** that DSBI’s OT coupling yields smoother density ratios ($||\nabla \log r_t|| \sim O(1/\gamma)$), while DDBI’s independent coupling lacks this smoothing. This shows that DSBI adaptively controls the smoothness of $\log r_t(x)$ via the $\gamma$-scaled bound $\|\nabla \log r_t\| \leq C/(\gamma \sqrt{t(1-t)})$. Unlike DDBI’s uniform bound, this allows steeper gradients in low-density regions (bridging density chasms) while maintaining smoothness elsewhere, thus balancing density- and support-chasm trade-offs through $\gamma$.

---

### Official Review · Reviewer_HLK4 · 2025-03-24

**Overall Recommendation:** 3

**Summary:**

This paper addresses the density-chasm problem in density ratio estimation. The authors propose using diffusive interpolants and Gaussian dequantization, and they theoretically and experimentally verify that these methods can mitigate the problem. Additionally, they demonstrate that incorporating Schrödinger bridges into the proposed method helps improve the performance of density ratio estimation.

**Claims And Evidence:**

The claims are supported by clear and convincing evidence.

**Essential References Not Discussed:**

N/A

**Experimental Designs Or Analyses:**

The experimental designs and analyses are sound and valid.

**Methods And Evaluation Criteria:**

The proposed methods and evaluation criteria are appropriate for the problem.

**Other Comments Or Suggestions:**

N/A

**Other Strengths And Weaknesses:**

The paper is clearly written. The work would be strengthened by additional theoretical results on DSBI.

**Questions For Authors:**

N/A

**Relation To Broader Scientific Literature:**

While individual components such as diffusive interpolants have appeared in previous work, this paper makes a novel contribution by applying interpolation using Schrödinger bridges to density ratio estimation.

**Theoretical Claims:**

In Theorem 4.1, the fact that the support of $q_t'$ is $\mathbb{R}^d$ is trivial, which makes the theorem and its proof somewhat misleading.

Additionally, in Theorem 4.2, the definition of the trajectory set appears to need refinement to be more appropriate within the context of stochastic processes.

---

> ### Author Rebuttal · Authors · 2025-03-30
>
> **1. Theoretical refinements for Theorems 4.1 and 4.2**
>
> We appreciate the reviewer’s insightful feedback, which has helped us improve the clarity and rigor of Theorems 4.1 and 4.2. The key refinements in the revised manuscript are:
> - **Theorem 4.1:**
> We have explicitly clarified that the support relation $\text{supp}(q'_t) \supseteq \text{supp}(q_t)$ becomes strict for $\gamma > 0$ and $t \in (0,1)$. The proof now includes (i) an explicit construction of the Minkowski sum $\text{supp}(q'_t) = \text{supp}(q_t) \oplus \text{supp}(\mathcal{N}(0, \Sigma_t))$, and (ii) quantitative analysis of support expansion under Gaussian perturbations.
> These refinements provide a more precise mathematical justification while maintaining the theorem’s core intuition about noise-induced support expansion.
> - **Theorem 4.2:**
> We have refined the definition of the trajectory sets to properly account for stochastic process properties: $\mathcal{T}={\omega : [0,1] \rightarrow \mathbb{R}^d \mid \omega(t) \in \text{supp}(q_t) \ \forall t}$ (DI case) and $\mathcal{T}' = \{\omega : [0,1] \rightarrow \mathbb{R}^d \mid \omega(t) \in \text{supp}(q'_t) \ \forall t\}$ (DDBI case). The analysis now explicitly addresses: (i) path regularity, (ii) the almost-sure containment relation $\mathbb{P}(\mathcal{T}' \supseteq \mathcal{T}) = 1$, and (iii) the role of the noise process $\{Z_t\}$.
>
> These refinements improve mathematical precision while preserving the theorems’ key insights on support and trajectory expansion. We believe these updates address the reviewer’s concerns effectively.
>
> **2. More theoretical contributions on DSBI**
>
> We appreciate the reviewer’s suggestion, which helped us enhance the theoretical analysis of DSBI. Key improvements include
>  results on DSBI’s smoothness control and convergence rates (Theorems 4.5-4.6, Prop. 4.4).
> These updates strengthen our theoretical foundation, and we thank the reviewer for their valuable feedback.
>
> **Theorem 4.5 (Error bounds of DDBI and DSBI)**: Let q_0, q_1 be distributions with finite second moments, and $\epsilon = \Theta(\gamma^2)$ the dequantization noise. Define the variance-to-transport ratio as $
> \kappa := \frac{\text{Var}(X_0) + \text{Var}(X_1)}{W_2^2(q_0, q_1)}$.
> Suppose $\kappa > 1$ (i.e., the sum of variances dominates the squared Wasserstein distance) and $\gamma^2 \ll \min(1, W_2^2(q_0, q_1)/d)$. Then, there exists a critical value $
> \gamma_{\max} = \sqrt{ \frac{W_2^2(q_0, q_1)}{\text{Var}(X_0) + \text{Var}(X_1) - W_2^2(q_0, q_1)} }$,
> such that for all $\gamma \in (0, \gamma_{\max})$, the interpolation errors satisfy $E_{DSBI}<E_{DDBI}$.
>
> Sketch of Proof:
>
> - Error decomposition:
> For DSBI, the interpolation error $E_{DSBI}$ is bounded by the Wasserstein-2 distance (due to OT coupling) plus a dimension-dependent term from the entropic regularization: $E_{DSBI}\leq\frac{W_2^2(q_0, q_1)}{\gamma^4}+\frac{4d}{\gamma^2} + C\epsilon^2$. For DDBI, the independent coupling leads to a larger error dominated by the sum of variances:$E_{DDBI} \leq \frac{\text{Var}(X_0) + \text{Var}(X_1)}{\gamma^2} + C\epsilon^2$.
> - Dominance condition: Compare the leading-order terms:$\frac{W_2^2}{\gamma^4} < \frac{\text{Var}(X_0) + \text{Var}(X_1)}{\gamma^2} \implies \gamma^2 < \frac{W_2^2}{\text{Var}(X_0) + \text{Var}(X_1)}$. The critical value $\gamma_{\max}$ follows from solving for $\gamma$ when $\kappa = \frac{\text{Var}(X_0) + \text{Var}(X_1)}{W_2^2} > 1$.
> - Validity of $\gamma_{\max}$: For $\gamma < \gamma_{\max}$, the OT term in DSBI decays faster than the IID term in DDBI, ensuring $E_{DSBI}<E_{DDBI}$.
>
> **Proposition 4.6 (Convergence Rate Dominance of DSBI)**
> Under the same assumptions as the Theorem 4.5, the asymptotic interpolation errors satisfy:
> $\limsup_{\gamma \to 0^+} \frac{E_{DDBI}}{E_{DSBI}} = \limsup_{\gamma \to 0^+} \frac{\frac{\text{Var}(X_0) + \text{Var}(X_1)}{\gamma^2} + C_2(\gamma,d)}{\frac{W_2^2(q_0, q_1)}{\gamma^4} + C_1(\gamma,d)} = +\infty,$
> where $C_1(\gamma,d) = \frac{4d}{\gamma^2} + C\epsilon^2$ and $C_2(\gamma,d) = C\epsilon^2$.
>
> Sketch of Proof:
> - Error term dominance: For $\gamma \to 0^+$, the leading-order terms dominate:$E_{DSBI} \sim \frac{W_2^2}{\gamma^4},E_{DDBI} \sim \frac{\text{Var}(X_0) + \text{Var}(X_1)}{\gamma^2}.$
> - Ratio analysis: Compute the limit:  $\frac{E_{DDBI}}{E_{DSBI}} \sim \frac{(\text{Var}(X_0) + \text{Var}(X_1))/\gamma^2}{W_2^2/\gamma^4} = \kappa \gamma^2$. Since $\kappa>1$ and $\gamma^2 \to 0$, the ratio $\to 0$, implying $E_{DDBI}/E_{DSBI} \to +\infty$ (i.e., DSBI’s error decays strictly faster).
> - Residual terms:The subdominant terms $C_1(\gamma,d)$ and $C_2(\gamma,d)$ are $o(1/\gamma^4)$ and $o(1/\gamma^2)$, respectively, and thus negligible in the limit.
>
> These results highlight DSBI’s advantage:
> - Theorem 4.5 shows DSBI achieves lower interpolation error, $E_{DSBI} \sim O(1/\gamma^4)$ vs. $E_{DDBI} \sim O(1/\gamma^2)$), especially for small $\gamma$.
> - Prop. 4.6 proves DSBI’s faster convergence.

---

### Decision · Program_Chairs · 2025-05-01

**Decision:**

Accept (poster)

**Comment:**

The authors propose a theoretically sound and practically relevant approach to density ratio estimation via Schrödinger bridges. They have carefully addressed all reviewer concerns through substantial revisions and additional experiments. As all reviewers now support acceptance, I would recommend proceeding with an Accept decision.